# Language Model Self-improvement by Reinforcement Learning Contemplation

**Jing-Cheng Pang**[1,2,*], **Pengyuan Wang**[1,2,*], **Kaiyuan Li**[1], **Xiong-Hui Chen**[1,2], **Jiacheng Xu**[1],
**Zongzhang Zhang**[1] **& Yang Yu**[1,2,3,◇]
[1] National Key Laboratory for Novel Software Technology, Nanjing University, China &
School of Artificial Intelligence, Nanjing University, China;
[2]Polixir.ai; [3] Peng Cheng Laboratory, Shenzhen, 518055, China

## Abstract

Language model self-improvement (LMSI) techniques have recently gained significant attention as they improve language models without requiring external supervision. A common approach is reinforcement learning from AI feedback (RLAIF), which trains a reward model based on AI preference data and employs a reinforcement learning algorithm to train the language model. However, RLAIF relies on the heuristic assumption that an AI model can provide effective feedback and correct wrong answers, requiring a solid capability of the language model. This paper presents a novel LMSI method, Reinforcement Learning Contemplation (RLC). We disclose that it is simpler for language models to evaluate a sentence than to generate it, even for small language models. Leveraging the gap between the evaluation and generation, RLC evaluates generated answers and updates language model parameters using reinforcement learning to maximize evaluation scores. Through testing on various challenging reasoning tasks and text summarization task, our experiments show that RLC effectively improves language model performance without external supervision, resulting in an answering accuracy increase ($31.23\% \rightarrow 37.09\%$) for BigBench-hard reasoning tasks, and a rise in BERTScore for CNN/Daily Mail summarization tasks. Furthermore, RLC can be applied to models of different sizes, showcasing its broad applicability.

## 1 Introduction

Scaling large language models (LLM) has led to remarkable performance in various natural language processing (NLP) tasks, such as language understanding, machine translation, and question answering (Zhao et al., 2023; Liu et al., 2023). However, training LLMs at scale often requires extensive high-quality supervised datasets (Huang et al., 2022a). To rectify this drawback, unsupervised training of LLMs has gained significant interest recently, offering the potential for training LLMs without relying on external supervision. This paper calls these methods *language model self-improvement* (LMSI) methods.

Two notable LMSI works are RLCAI (Bai et al., 2022) and RLAIF (Lee et al., 2023). RLCAI employs a model to evaluate which of the two answers better adheres to a set of rules or principles, subsequently creating a dataset of AI preferences. The system then trains using reinforcement learning (RL), with the preference model as the reward signal. This method aims to enhance specific properties of generated text, such as helpfulness and harmlessness. RLAIF, on the other hand, follows a similar training process (i.e., reward model + RL) but further validates that AI feedback performs comparably to human feedback in summarization tasks. However, these approaches rely on the heuristic assumption that AI models can provide effective feedback or correct errors, which requires a strong capability of the language models and is not sufficiently justified in their works. Besides, the need to collect preference data and train an extra reward model introduces added costs.

Compared to these methods, humans possess the capacity for self-improvement through reflection on past experiences, as indicated by contemplative science within the field of human psychology (Van Gordon et al., 2022). This process does not necessitate the explicit learning of an additional pref-

---

*Equal contribution. ◇ Corresponding: yuy@nju.edu.cn.

erence model for self-feedback. Consequently, we aim to investigate the analogous self-improvement capabilities within LLM. In this paper, we propose a novel LMSI method, which is grounded on the observation that it is simpler for a language model to evaluate a sentence than to generate it. For example, while writing an attractive story can be challenging, identifying the generated text is relatively easy. Fig. 1 illustrates the disparity between text generation and evaluation. We comprehensively investigate the text evaluation ability of language models through experiments across various NLP tasks. Leveraging this evaluation ability, we present **R**einforcement **L**earning **C**ontemplation (RLC). Given an unlabelled question set, the language model generates answers to unlabeled samples and directly evaluates the quality of these answers. The model is then updated using reinforcement learning to optimize for maximum evaluation scores. Our proposed method employs self-evaluation results as rewards and utilizes reinforcement learning to optimize the language model. We refer to this process as *reinforcement learning contemplation*.

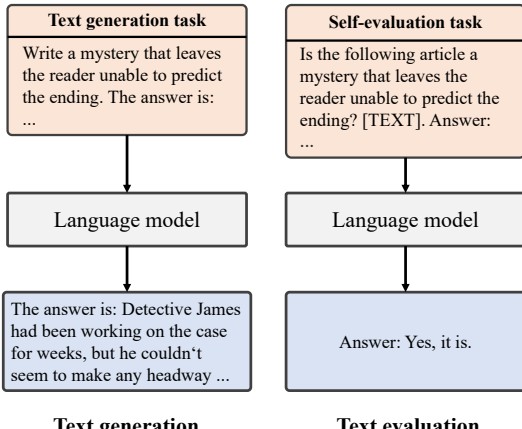

Figure 1: A comparison between text generation and evaluation. Text evaluation involves evaluating and analyzing certain properties of existing text, while generation requires the creation of entirely new text.

The contribution of this work can be summarized as follows: This work explains why the RLAIF functions effectively. We provide evidence that evaluation tasks are simpler than generation tasks for LLMs. This difference can be leveraged to boost the performance of LLMs of various sizes. To our knowledge, this is the first work that presents the gap between text generation and evaluation. Besides, we propose a novel LMSI method for unsupervised training of LLMs, eliminating the need for external supervision and training reward models. Finally, we conduct comprehensive experiments to demonstrate that LLM, trained with our method, can self-improve its ability in solving reasoning and text summarization problems. We also present that RLC can be applied to LLMs with a parameter range of 80M to 780M, and the trained LLM generalizes well to new and unseen datasets, demonstrating the extensive applicability of the proposed method.

## 2 RELATED WORK

**Language model self-improvement.** LMSI methods have drawn significant attention recently. These methods can be broadly classified into two categories, namely *noninvasive* and *invasive* methods. Noninvasive methods utilize AI feedback to guide the LLM to revise its output during the generation process. This approach involves a repeated refinement and correction of the model's output, based on the LLM self-evaluation (Huang et al., 2022a; Cobbe et al., 2021; Weng et al., 2022; Raman et al., 2022). For example, the re-prompting (Raman et al., 2022) method identifies errors in the current plan generated by the LLM and revises the current step if an error is detected. ReAct (Yao et al., 2022) and Self-Refine (Madaan et al., 2023) provides specific design to correct the self-generated error using language models. On the other hand, invasive methods update the LLM parameters using AI feedback as reward signals. Recent advancements in this field include RLAIF (Lee et al., 2023), which trains a reward model based on AI preference data and use a reinforcement learning algorithm to train the LLM. In contrast prior studies relied mostly on comprehensive experiment results, this work explains why such a self-evaluation mechanism work well: we provide evidence that evaluation tasks are simpler than generation tasks for LLMs, and this difference can be leveraged to boost the performance of LLMs of various sizes.

**Train language model with unlabeled data.** Learning from unlabelled data is a promising approach for enhancing the model efficiency. Self-training is a popular technique in this field, which entails assigning pseudo labels from a learned classifier to unlabeled data. These pseudo-labeled examples are then utilized to enhance the initial model training (RoyChowdhury et al., 2019; Chen et al., 2021; He et al., 2020). In recent years (Huang et al., 2022a), self-training has been employed to fine-tune large-scale language models such as PaLM (Chowdhery et al., 2022) with 540B parameters, and it

has shown immense potential. However, it fails to present efficiency on smaller size of LLMs. In contrast to methods that utilize LLMs to generate pseudo-labels, our approach employs RL to train LLMs, which has proven more effective than supervised learning (Stiennon et al., 2020).

## 3 PRELIMINARY

We begin with a vocabulary $\Sigma$ and an LLM $\mathcal{M}$ which takes a token sequence of the question $q = \{q_0, ..., q_n\}$ as input and predicts the next token using autoregressive modeling: $o_{t+1} = \mathcal{M}(q, \{o_0, \ldots, o_t\})$, where $q \in \Sigma^n$ and $o_t \in \Sigma$. To fine-tune LLM with RL, we can view this problem by Markov Decision Process (MDP) (Puterman, 1994; Sutton & Barto, 1998), which is described as a tuple $(\mathcal{S}, \mathcal{A}, P, \mathcal{R}, \gamma, d_0)$:

- State space $\mathcal{S}$: the space of input token sequences $q \cup \{o_0, o_1, \ldots, o_t\}$. $s_0$ is defined as the token sequence of question $q$.
- Action space $\mathcal{A}$: the space of tokens $o_t$.
- Reward function $\mathcal{R}(q, \{o_0, \ldots, o_t\})$: a score that reflects the quality of the generated answer to the question, which can be obtained from human feedback or a predefined criterion. The reward is typically given when the complete answer has been generated.
- Transition $P$: $s_{t+1} = s_t \cup o_{t+1}$.
- Initial distribution $d_0$: the distribution of question $q$.

Here, the LLM $\mathcal{M}$ acts as a policy mapping from state space to the probability space over action space. The objective of RL is to train the policy to maximize the expected return:

$$\mathbb{E}\left[\sum_{t=0}^{\infty} \gamma^t \mathcal{R}(q, o) \big| q \sim d_0, o_t \sim \mathcal{M}(\cdot|s_t)\right]. \tag{1}$$

In RLC, we formulate the LLM text generation problem as an MDP, analogous to the one described above, with the reward function derived from evaluation results. To simplify the notation, we use $o \sim \mathcal{M}(q)$ to represent the autoregressive sampling of a complete answer $o$ from the language model $\mathcal{M}$, based on the input question $q$.

## 4 EVALUATION IS SIMPLER THAN GENERATION

In this section, we aim to verify the text evaluation ability of LLMs by investigating three key topics in subsequent subsections: (1) comparison of LLMs' ability in text generation and evaluation; (2) the correlation between evaluation results and the established evaluation metrics; and (3) potential for using text evaluation to improve LLMs.

### 4.1 COMPARISON OF EVALUATION AND GENERATION

We conduct experiments to compare the text generation and evaluation abilities of LLMs using the CommonGen (Lin et al., 2020) task, which involves generating a sentence that describes an everyday scenario based on a given set of common concepts such as *{dog, frisbee, catch, and throw}*. Specifically, we use FLAN-T5 (Chung et al., 2022) as the LLM to generate text based on common conceptual questions provided by CommonGen and evaluate the accuracy of the generated text through human evaluation. In addition, we assess the LLM's text evaluation ability by using the same LLM to evaluate whether the generated text meets the given concepts. Appendix B shows more experiment details (e.g., the prompts we use). As the experiment results presented in Fig. 2, we observe that the evaluation accuracy exceeds the generation accuracy in all scales of models. Especially when the parameter size of the model is small (see FLAN-T5-Large/XL), the evaluation accuracy significantly outperforms the text generation by 15%. These results indicate that it is simpler to self-evaluate the generated text than to generate high-quality text that meets contextual requirements. However, it is essential to note that the evaluation accuracy is affected by the quality of the target of evaluation, and this experiment is only a preliminary demonstration of the LLM's ability to self-evaluate. We conduct more experiments to verify the text evaluation ability of LLM further, as presented in the following subsections.

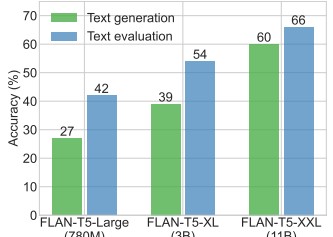

Figure 2: Comparison of the text generation and evaluation.

| Task \ Metric | BLEU | ROUGE | BERTScore |
|---|---|---|---|
| CNN/Daily Mail | 0.26 | 0.16 | 0.23 |
| IWSLT 2017 | 0.21 | 0.28 | 0.29 |

Table 1: The correlation coefficient between evaluation and different metrics. The evaluation result correlates positively with all three metrics. The experiments are conducted with FLAN-T5-XL.

### 4.2 CORRELATION BETWEEN EVALUATION AND ESTABLISHED METRICS

This section provides an analysis of the correlation between text evaluation and established metrics using two datasets: the CNN/Daily Mail dataset (Nallapati et al., 2016) for text summarization and the IWSLT 2017 dataset (Cettolo et al., 2017) for text translation. To find the relationship between evaluation and established metrics, LLM generates two answers for each question in these tasks and then self-evaluates to determine which answer is better. Additionally, we utilize three metrics, namely BLEU (Papineni et al., 2002), ROUGE (Lin, 2004), and BERTScore (Zhang et al., 2020), to compare the two answers, calculate their correlation coefficient with evaluation, and present the findings in Tab. 1. As demonstrated by the results, the evaluation exhibits a positive correlation with all the considered metrics, indicating a consistent measurement of the quality of the generated text, and it is a reliable way to measure the quality of the generated text.

### 4.3 POTENTIAL FOR SELF-IMPROVEMENT

The previous section shows a positive correlation between evaluation and established metrics. However, it remains to be seen whether and how the text evaluation can be utilized to improve text generation. To investigate this problem, we design a text generation strategy that utilizes the text evaluation ability of LLM: the LLM re-samples the answer based on the self-evaluation result, which will be explained in the following paragraphs. We conduct experiments on two NLP benchmarks.

One such benchmark is Bigbench-hard (Srivastava et al., 2022), which includes multiple reasoning tasks consisting of multiple-choice and direct-generation tasks. We compare two answer generation strategies: (1) **w/o SE**: the answer is a directly deterministic output of the LLM, and (2) **w/ SE**: LLM generates an answer and evaluates its correctness. If the evaluation indicates the answer is incorrect, LLM re-generates an answer as the final output. We tested these two strategies on multiple Bigbench-hard tasks, and the results are presented in Tab. 2. The experiment results demonstrate that with evaluation, the answer accuracy outperforms or is comparable to that of direct answer generation on 10 of 12 evaluation tasks. This result justifies using evaluation to help LLM improve answer accuracy.

| | Reasoning about Colored Objects | Logical Deduction (7) | Tracking Shuffled Objects (5) | Object Counting | Tracking Shuffled Objects (3) | Geometric Shapes |
|---|---|---|---|---|---|---|
| w/o SE | 43.2% | 30.3% | **13.7%** | **22.6%** | 25.7% | 9.6% |
| w/ SE | **44.7%** | **34.5%** | 10.6% | 22.5% | **28.5%** | **10.5%** |

| | Web of Lies | Sports Understanding | Logical Deduction (3) | Logical Deduction (5) | Penguins in a Table | Navigate |
|---|---|---|---|---|---|---|
| w/o SE | 50.5% | **55.1%** | 50.7% | 29.7% | 30.8% | **51.1%** |
| w/ SE | **51.0%** | **55.1%** | **57.2%** | **34.8%** | **34.0%** | 49.9% |

Table 2: Comparison of the answer accuracy between answer generation with/without self-evaluation. Full results on all 27 BigBench tasks are presented in Appendix C.3.

Furthermore, we also conduct experiments on two text summarization tasks, CNN/Daily Mail and BBC (Gupta et al., 2022). As it is not meaningful to evaluate the correctness of generated summa-

rizations, we use a different approach to utilize self-evaluation ability of LLM in this experiment: (1) **w/ SE**: LLM samples three different answers and evaluates which answer is the best one as the final answer, and (2) **w/o SE**: LLM samples three different answers, and we present the average score of the three answers. As shown in Fig. 3, the generated answers have higher scores under all three metrics when self-evaluation is used. This result suggests that evaluation can potentially improve the quality of the generated text, which serves as a stepping stone for building RLC method.

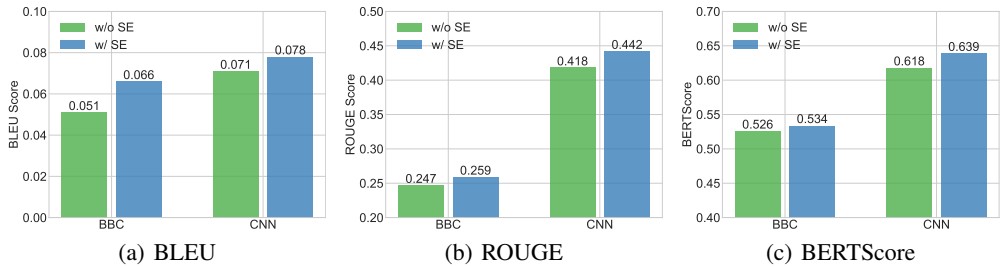

|     |     |     |
| --- | --- | --- |
| (a) BLEU | (b) ROUGE | (c) BERTScore |

Figure 3: Comparison of text generation with/without self-evaluation on text summarization tasks.

# 5   SELF-IMPROVEMENT BY REINFORCEMENT LEARNING CONTEMPLATION

In the previous section, we observed a gap between text evaluation and generation for LLM, and present an opportunity to enhance the generation abilities of LLMs. For example, LLMs can effectively evaluate the sentences they generate. These evaluation results serve as learning signals to guide the LLMs towards improved generation ability.

In this section, we will elaborate on our proposed LMSI method, RLC, that improves LLM ability without external labels based on self-evaluation. We are given a pre-trained language model $\mathcal{M}$ and an unlabeled training dataset $\mathcal{D}^{\text{train}} = \{q_i\}_{i=1}^{|\mathcal{D}|}$. The overview of our method is illustrated in Fig. 4, which iterates through the following steps:

- Gather question-answer pair $(q, o)$ via $q \sim \mathcal{D}^{\text{train}}$, $o \sim \mathcal{M}$.
- Evaluate the question-answer and obtains the reward $r$ via Eq. 2.
- Self-improvement with reinforcement learning.

**Answer generation to unlabeled questions.** RLC randomly samples a batch of questions from $\mathcal{D}^{\text{train}}$ and asks the LLM to generate answers to these questions. For different types of questions, we use different prompts to generate answers, as described in Appendix B.3. For reasoning problems, we use the Chain-of-Thought (CoT) prompt Wei et al. (2022), which has been shown to effectively improve LLM performance in previous studies (Wang et al., 2023; Huang et al., 2022a).

**Self-evaluation as the reward.** After gathering the question-answer pair $(q, o)$, RLC self-evaluate the generated answer to produce the reward for RL training:

$$R(q, o) = \phi(\mathcal{M}(p_{\text{EP}}, q, o)), \tag{2}$$

where $\phi$ is a processing function that processes the LLM output to the numerical reward and $p_{\text{EP}}$ is the prompt used for self-evaluation. Compared to previous works that refine/revise the generated text based on the evaluation results, RLC uses an *invasive learning style*, i.e., updating the language model to improve its generation ability. The advantage of such invasive training is the quick response speed of the trained model: it does not need to repeatedly generate and revise the answer.

RLC uses two types of prompts for evaluating the self-generated text: (1) Correctness Evaluation Prompt (**CEP**): *"Is the answer to the question correct? The question is: [Q]. The answer is: [A]"*, and (2) Quality Evaluation Prompt (**QEP**): *"Please evaluate the answer to the question and give me an evaluation score from 1 to 10. The question is: [Q]. The answer is [A]"*. Depending on the type of question, either CEP or QEP is utilized to evaluate the generated text.

RLC applies CEP to assess the factual accuracy of the generated text, which is essential for tasks such as question-answering and reasoning. The CEP prompts LLMs to verify the answer's correctness and

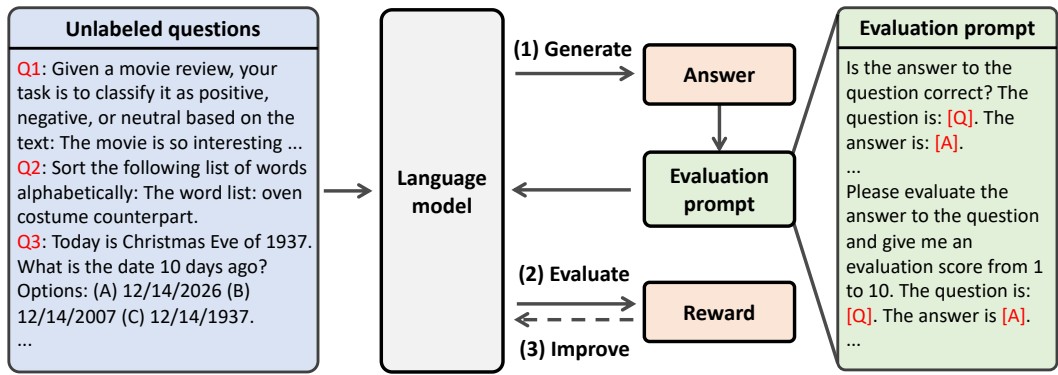

Figure 4: Overall training procedure of RLC, which iterates through three steps: (1) Answer generation to the unlabeled questions. (2) Self-evaluation by asking LM using *evaluation prompt*, with the evaluation results as the reward. (3) Update the language model to maximize the reward using reinforcement learning algorithms. The solid lines represent the data flow, while the dashed line represents the update of LLM parameters.

identify any factual errors. Given a question $q$ and the answer $o$ generated by the LLM, the reward $R(q, o)$ is a 0-1 value based on the evaluation result. RLC employs QEP to evaluate the overall effectiveness and coherence of the generated text in terms of its writing, structure, and style. Quality is often evaluated on a scale, with higher scores indicating the text presents better effectiveness and fits some given requirements more. The QEP prompts LLMs to rate the text's quality on a scale of 1 to 10 based on how well it answers the question. This prompt type helps solve generation tasks like translation and summarization.

During training, the reward distribution may change as the LLM is updated. We use the initial pre-trained LLM $\mathcal{M}^*$ for self-evaluation while keeping it fixed to ensure stable training.

**Self-improvement through reinforcement learning.** With the evaluation reward, the LLM $\mathcal{M}$ can be updated using any RL algorithm to maximize the reward. In our case, we employ the Proximal Policy Optimization (PPO) algorithm (Schulman et al., 2017), which has demonstrated promising results in applications (Huang et al., 2022b). To ensure better exploration, we apply entropy regularization, which prevents the sampling phase from converging too early. Additionally, we use the Kullback-Leibler (KL) divergence to prevent $\mathcal{M}$ from deviating too far from the initial pre-trained LLM.

## 6 EXPERIMENTS

We conduct a series of experiments to demonstrate the effectiveness of our proposed RLC method. Our experiments encompass the following topics: (1) comparison of the proposed method with baseline methods on various NLP tasks (Section 6.2); (2) the applicability of the proposed method to different model sizes (Section 6.3); and (3) generalization on unseen tasks (Section 6.4). We first introduce our experimental setup.

### 6.1 EXPERIMENT SETUP

**Dataset for evaluation.** We use the BigBench (Srivastava et al., 2022) benchmark to conduct our experiments. BigBench is a challenging reasoning task requiring the language models' complex reasoning capabilities. The tasks in BigBench are pretty diverse, including reasoning the final results of a sequence of actions, understanding dates, and completing tasks that require simple arithmetic calculations. In our experiments, we use 12 challenging tasks from the BigBench-Hard datasets[1], which covers judgments, multiple choices, and text generation tasks.

**Baselines for comparison.** We compare RLC with representative baselines that improve LLM without supervised data. These methods include: (1) **RLAIF** trains a reward model based on AI

---

[1]Detailed descriptions about the tasks are in `https://github.com/google/BIG-bench/blob/main`.

|  | Example inputs | Example outputs |
|---|---|---|
| Judgement | "Jamey lies. Vernell says Jamey tells the truth. Millicent says Vernell lies. Rashida says Millicent . . . Does Vina tell the truth?" | "Vernell is lying because Vernell is telling . . . The answer: Yes." |
| Text generation | "I have a chicken, a snail, three dogs, a snake, and a goat. How many animals do I have?" | ". . . So I have a chicken + snail + four dogs + a snake + a goat = 7 animals. So the answer is 7." |
| MultiChoice | "Here is a table where the first line is a header and each subsequent line is a penguin: name, age, . . . How many penguins are there in the tables? Options: (A) 1 (B) 2 (C) 3 (D) 4 (E) 5." | "There are 4 penguins in the first table and 4 giraffes in the second table. The answer: (D)." |

Table 3: Examples of inputs and outputs for the tasks in our experiments.

preference data and employs a RL algorithm to train LLM; (2) Reinforcement Learning Fine-Tuning (**RLFT**) fine-tunes the LLM using reinforcement learning (RL) and employs oracle metrics as the reward. The metric used is answer accuracy for multiple-choice, judgement, and fill-in-the-blank tasks. For summarization tasks, BERTScore is used as the metric; (3) **Self-train** (Huang et al., 2022a) utilizes the SC method to generate "high-confidence" answers for unlabelled questions as the pseudo-labels. It then fine-tunes the LLM on the generated data with supervised learning; (4) Direct Generation (**DG**) directly generates the answer with the deterministic output of the LLM; (5) Self-consistency (**SC**) (Wang et al., 2023) samples a diverse set of reasoning paths instead of solely relying on the greedy LLM output. It then selects the most consistent answer by marginalizing out the sampled reasoning paths. In our experiments, we use the number of sample paths as 40; (6) **Self-refine** (Madaan et al., 2023) employs a LLM to generate an initial output. This output is evaluated from multiple perspectives and revised by the same LLM. (7) **Best-of-N** (OpenAI, 2022) generates N answers and uses the same model to evaluate and choose the best answer. Some may concern about the fairness of the comparison, as Best-of-N and SC generates multiple answers and selects one best answer from the answer candidates, while other method only generates one answer as the final answer. It is important to highlight that our experimental focus lies in assessing the performance gains of various methods in enhancing language models. Our evaluation primarily measures the performance disparity between DG (which generates one answer during the generation process).

**Implementation details.** We utilize PPO to train the LLM for 6,000 gradient steps for each task, with a batch size of 12. The PPO implementation is from the trlx repository on GitHub (CarperAI, 2020). We implement RLC using CEP in reasoning tasks and QEP for summarization task. Unless otherwise specified, we use FLAN-T5-Large, which has 780M parameters, as our LLM in the experiments. All reported results are averaged over three random trials, except for RLAIF with one seed, and the experiments can be conducted using two GTX 3090 graphics cards with 24GB of memory. We provide specific hyper-parameters and more detailed implementation descriptions in Appendix B. All RL-based methods use the same hyper-parameters for training RL algorithms. In our experiments, we consistently employ the same prompts for all baseline models and the RLC. For each method, we utilize the CoT prompts, specifically, "Let's think step by step." A comprehensive list of the prompts used in our experiments can be found in Table 7 of the Appendix.

## 6.2 MAIN RESULTS

**Results on BigBench**. We evaluate RLC extensively using the BigBench dataset. Tab. 4 presents the answer accuracy of the LLMs trained with RLC and other baseline methods. We see RLC outperforms all the baselines without the supervised label. In particular, the RLC improve answer accuracy over DG from 31.23% to 37.09%. On some tasks (e.g., Reasoning about Colored Objects), RLC even catches up with the performance of RLFT, which fine-tunes LLM with Oracle supervision. This result

could be attributed to the fact that the RLC effectively finds the proper direction for improvement, contributing to the significant improvement of LLM. Besides, RLC outperforms all baselines on most tasks, demonstrating the effectiveness of the proposed method. SC presents good performance on these tasks and achieves the highest score on 3 of 12 tasks. It is important to note that SC is also compatible with RLC, which can be used to improve the algorithm performance further. Besides, RLC can achieve comparable or better performance than Best-of-N and self-refine, while RLC takes less computation time during the generation process. We also compare with RLAIF (with GPT2 (Radford et al., 2019) as reward model) with RLC. The experiment results show that RLC achieve a better performance. This can be attributed that reward model with smaller size may not provide effective preference results. The training curves are depicted in Fig. 7, located in the Appendix C.1.

| | Reasoning about Colored Objects | Logical Deduction (7) | Tracking Shuffled Objects (5) | Object Counting | Tracking Shuffled Objects (3) | Geometric Shapes |
|---|---|---|---|---|---|---|
| RLFT | 32.1% | 45.7% | 12.4% | 42.6% | 33.6% | 18.9% |
| DG | 32.0% | 35.2% | 12.4% | 31.9% | 31.2% | 5.2% |
| SC | **39.6%** | 27.6% | 12.4% | 24.0% | 33.6% | 15.6% |
| Self-train | 19.5% | 13.1% | **15.5%** | 11.7% | 33.1% | 12.4% |
| Self-refine | 25.2% | 13.2% | 8.0% | 18.0% | 25.2% | 10.0% |
| Best-of-N | 26.8% | 12.8% | 12.1% | 14.0% | 30.0% | 8.4% |
| RLAIF | 30.4% | 36.9% | 11.4% | 32.5% | 32.8% | 14.0% |
| RLC | 35.0% | **39.2%** | 12.2% | **35.4%** | **33.6%** | **17.8%** |

| | Web of Lies | Sports Understanding | Logical Deduction (3) | Logical Deduction (5) | Penguins in a Table | Navigate |
|---|---|---|---|---|---|---|
| RLFT | 72.2% | 68.8% | 58.6% | 41.9% | 44.2% | 55.6% |
| DG | 43.6% | 53.2% | 39.6% | 28.4% | 15.7% | 46.4% |
| SC | 48.4% | 53.6% | 42.8% | 30.8% | **35.2%** | **62.8%** |
| Self-train | 51.1% | 51.1% | 34.0% | 18.4% | 19.7% | 48.7% |
| Self-refine | 47.2% | 50.0% | 28.4% | 17.2% | 17.8% | 46.0% |
| Best-of-N | 50.0% | **59.2%** | 42.0% | 22.0% | 17.8% | 45.2% |
| RLAIF | 52.1% | 56.1% | 22.0% | 33.7% | 19.8% | 48.8% |
| RLC | **52.9%** | 53.5% | **44.0%** | **34.6%** | 29.8% | 57.1% |

Table 4: The answer accuracy of RLC and baseline methods on the Bigbench-hard tasks. Each value represents the average answer accuracy of the last three training iterations. The highest performing value among methods without external labels is highlighted in **bold**.

**Results on summarization task.** In addition to reasoning tasks, we evaluate the performance of RLC on a challenging text generation benchmark: CNN/Daily Mail. Self-train method is implemented as follows: We utilize the initial LLM to generate the summarization label to the text in the dataset, after which the LLM is fine-tuned on this newly generated data. As illustrated in Fig. 5, RLC enhances the BERTScore from 0.886 to 0.899 in the summarization task, while self-train hardly improves the model performance on this dataset. Unlike the BigBench tasks, where the generated text is relatively short, the LLM is required to produce longer text for these two tasks. The experimental results demonstrate that RLC effectively improves the text generation capabilities of the LLM.

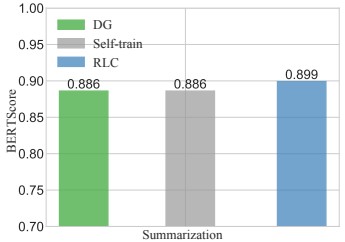

Figure 5: Performance on summarization task.

## 6.3 EVALUATION ON MODEL SIZE VARIATIONS

We conduct experiments to assess the performance of RLC across various language model sizes. We select three distinct models: FLAN-T5-Small, FLAN-T5-Base, and FLAN-T5-Large, containing 80M, 250M, and 780M parameters, respectively. We train these models using RLC on three challenging BigBench tasks, with the final scores presented in Fig. 6. In general, RLC effectively enhances performance across different scales of language models. Notably, when the parameter size is small (80M) and the base score is low, the language model exhibits a significant improvement.

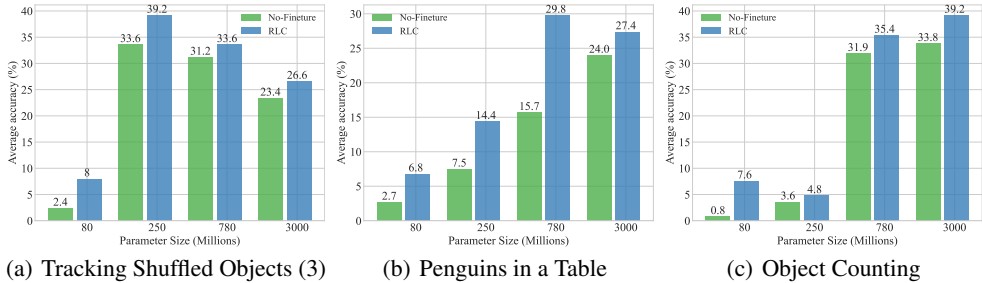

(a) Tracking Shuffled Objects (3)     (b) Penguins in a Table     (c) Object Counting

Figure 6: Performance of RLC on different sizes of language models. RLC can improve the answer accuracy over the initial pre-trained model without supervision.

### 6.4 GENERALIZATION TO UNSEEN TASKS

The primary objective of an LMSI method is to facilitate scalable training and enable the language model to improve continuously. In this section, we explore the potential for continuous self-improvement of the LLM using the RLC. The ability to generalize to unseen tasks is a crucial aspect of scalability. Thus, we assess whether the improved generative capability can generalize to datasets previously unseen.

Specifically, we utilize RLC to simultaneously train an LLM on five tasks, namely Geometric Shapes, Logical Deduction (3), Logical Deduction (5), Navigate and Tracking Shuffled Objects (3). After that, we test the trained LLM on five new tasks. As the evaluation results depicted in Tab. 5, The increase in performance is not necessarily correlated with the similarity between the training tasks and the evaluation tasks. For instance, Logical Deduction (3) (+1.5) and Tracking Shuffled Objects (5) (-0.1) both have tasks akin to the training tasks. It is important to note that the tasks in question were not seen during the training process. Despite this, the RLC method demonstrates an overall improvement in performance on average. These findings underscore the potential of RLC for application across a more comprehensive array of datasets, thereby augmenting the overall performance of LLMs.

| | Logical Deduction (7) | Object Counting | Penguins in a Table | Sports Understanding | Tracking Shuffled Objects (5) | Average |
|---|---|---|---|---|---|---|
| Acc. | 36.7 (**+1.5**) | 32.7 (**+0.7**) | 18 (**+2.2**) | 52.8 (-0.4) | 12.3 (-0.1) | **30.5 (+0.8)** |

Table 5: The answer accuracy on five unseen datasets.

### 7 CONCLUSION

This paper introduces a novel approach to enable LLM to self-improve: reinforcement learning contemplation (RLC). We demonstrate the performance gap of generation and evaluation of LLMs and the potential for evaluation boosting generation through extensive experimental analysis. By utilizing evaluation ability, LLMs can obtain valuable insights into their performance and identify areas for improvement. Our experiments demonstrate that RLC effectively improves LLM performance across various text generation tasks. Moreover, we illustrate the potential for the RLC method to scale with varying model sizes and training data. However, some things could be improved in our approach. First, like previous unsupervised methods for training language models, RLC requires an unlabeled dataset to generate answers and facilitate self-improvement. It would be worthwhile to explore training LLMs without relying on the datasets. For instance, through self-evaluation, an LLM can improve its specialized capabilities in text generation, such as outputting positive reviews. Additionally, our current implementation utilizes two separate models for evaluation and improvement. It remains to be investigated how the evaluation ability of the LLM will also change as it gets updated. Lastly, we primarily evaluate RLC on LLMs with 780M parameters due to computational limitation. Future experiments should include larger language models to better demonstrate the method's application range. We hope this work could provide new insights into LMSI. Future research could address the above questions to develop more efficient and effective training methods for language models.

ACKNOWLEDGEMENT

This work was supported by the National Science Foundation of China (61921006), the Major Key Project of PCL (PCL2021A12). The authors would like to thank Lei Yuan for discussing the method details, and anonymous reviewers & chairs for providing value comments.

SOCIETAL IMPACT

The introduction of RLC offers a promising solution to the challenges associated with fine-tuning large language models using external labels. This approach can reduce the manual cost and time required for providing supervision, thereby increasing access to NLP technologies for individuals and organizations with limited resources. Moreover, the enhanced accuracy and effectiveness of LLMs through the use of RLC have implications for a wide range of NLP tasks, including machine translation, reasoning problems, and other text generation tasks. As a result, RLC potentially influences diverse industries such as healthcare, finance, and education, where an accurate and reliable language model is essential. RLC present an exciting advancement in the field of NLP that could impact society, while we must ensure that the self-improvement direction of the LLMs remains aligned with human values.

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

# Appendix

## Table of Contents

## A  DISCUSSION

### A.1  THE USAGE OF UNLABELLED QUESTIONS DATASET

In Section 5, we introduce RLC, assuming the availability of a training dataset $\mathcal{D}^{\text{train}}$ consisting of unlabeled questions. This raises two questions: (1) what data type does the dataset contain? and (2) does RLC work without an unlabeled dataset?

Concerning the first question, the training dataset in our experiments involves questions in $\mathcal{D}^{\text{train}}$ that adhere to an *instruction + problem* format. For example, a question might be "Summarize the following article (instruction). [TEXT] (problem)". In this way, RLC can train an LLM using any open-source datasets in the NLP community by appending instructional text to the problems within these datasets. Tab. 6 presents the instructions we used in our experiments, where '[TEXT]' denotes the questions/text in the original dataset. This way of using a dataset is similar to instruction learning Lou et al. (2023), which utilizes task instructions to guide the learning system in rapidly adapting to various new tasks. We posit that as the data in $\mathcal{D}^{\text{train}}$ becomes more diverse and abundant, RLC can be employed to train an LLM with generalization ability. Our experiments in Section 6.3 offer preliminary validation of this outcome.

| Task | Instruction |
|---|---|
| BigBench-hard | [TEXT] Let's think step by step. |
| Text summarization | Please give a summary of the following text. Text: [TEXT] Answer: |

Table 6: Instructions for different tasks in our experiments.

As for the second question, although our experiments utilize an unlabeled dataset to train the LLM, we are interested in exploring whether RLC can be applied without a dataset. To achieve this, we need an objective, which serves as an evaluation criterion for assessing the quality of the generated text, to train the language model. For example, we might expect an LLM to consistently generate positive or polite text. In such a case, we can prompt the LLM to generate text randomly from the start token and evaluate its positivity or politeness using either the CEP or the QEP. This approach allows the LLM to optimize toward the desired attribute. However, it is essential to note that some training objectives or text attributes might be challenging to evaluate, making them unsuitable for self-improvement using the reinforcement learning contemplation method.

### A.2  SAMPLING STRATEGY

In our experiments, RLC employs a random sampling approach to select questions from the dataset. Although this method is effective, alternative sampling techniques could enhance the process. For instance, prioritizing questions with more uncertainty could be beneficial. The field of active learning offers many practical sampling methods (Xu et al., 2013) that could potentially augment learning efficiency. We posit that these active sampling strategies align well with the concept of *self-improvement*. However, in this study, our primary focus lies in the design of the LMSI method, for which we maintain the use of random sampling.

## B  EXPERIMENT DETAILS

In our experiments, we utilize the open-sourced RL repository, trlx, to implement the reinforcement learning contemplation. All experiments are run three times with different random seeds. This section will present more experiment details omitted in the main body due to the space limitation, including the evaluation task we use, the baselines, the prompts in different experiments, the hyper-parameters for reproducibility, etc.

B.1 TASKS FOR EVALUATION

In our experiments, we use five challenging benchmarks in the NLP domain to conduct various experiments to support our method. This section gives a detailed introduction to these benchmarks, which are omitted in the main body.

**CommonGen** Lin et al. (2020) is a task that focuses on constrained text generation and includes a benchmark dataset. Its primary objective is to assess a machine's ability to generate common sense reasoning. The task requires constructing a coherent sentence that describes everyday scenarios using a predefined set of concepts. CommonGen presents significant challenges, as it demands two primary skills: (1) utilizing background knowledge for relational reasoning and (2) effectively managing combination generalization of concealed concept combinations. We employ CommonGen to evaluate the text generation and self-evaluation ability of LLM, wherein the LLM is tasked with generating a sentence based on four concepts simultaneously.

**BigBench-hard** Srivastava et al. (2022) consists of 27 challenging tasks designed to evaluate the reasoning abilities of language models. These tasks present increased difficulty due to their complexity. In our experiments, we employ 12 challenging tasks that focus on evaluating different aspects of reasoning ability:

- Reasoning about Colored Objects (multiple choices): Answer simple questions about the colors of objects on a surface.
- Logical Deduction (multiple choices): Deduce the order of a sequence of objects.
- Tracking Shuffled Objects (multiple choices): Determine the final positions of a set of objects given their initial positions and a description of a sequence of swaps.
- Object Counting (text generation): Questions that involve enumerating objects of different types and asking the model to count them.
- Geometric Shapes (text generation): Name geometric shapes from their SVG paths.
- Web of Lies (judgement): Evaluate a random boolean function expressed as a word problem.
- Sports Understanding (judgement): Determine whether an artificially constructed sentence relating to sports is plausible or implausible.
- Penguins in a Table (text generation): Answer questions about a table of penguins and their attributes.
- Navigate (judgement): Given a series of navigation instructions, determine whether one would end up back at the starting point.

Note that specific tasks (e.g., Logical Deduction) encompass the same topic but are presented at varying difficulty levels, and we introduce these tasks with varying levels together.

The **CNN/Daily Mail** Nallapati et al. (2016) dataset serves as a widely recognized benchmark for text summarization. Comprising over 300,000 news articles from CNN and the Daily Mail and corresponding human-written summaries, this dataset has been instrumental in training and evaluating various text summarization models, including extractive and abstractive methods. Each article features several highlights summarizing its main points, making the dataset ideal for training and testing automatic text summarization models that aim to generate concise versions of the original text while retaining crucial information.

The **BBC** Gupta et al. (2022) dataset, created by the BBC News website, is another widely used resource for text summarization. The dataset has been pre-processed and annotated with human-generated summaries, encompassing approximately 2,225 news articles on diverse topics such as politics, entertainment, technology, and sports. These concise summaries, typically 3 to 4 long sentences, make the dataset invaluable for training and evaluating text summarization models.

B.2 BASELINES

In our experiments, we compare RLC with representative baseline methods. This section presents these baselines and their implementation details.

(1) SC: Self-Consistency. SC enhances the LLM's answers without updating its parameters. Its motivation is similar to ensemble methods, which sample multiple answers simultaneously and vote among them to select the most consistent one. SC has demonstrated significant capabilities in reasoning problems. In our experiments, the sampling path (i.e., the number of answers that LLM samples) is set to 40.

(2) Self-train: Language Model Self-train. LMSI combines the SC method with supervised learning fine-tuning. It first uses SC to generate answers for all questions in the dataset and then fine-tunes the LLM using supervised learning. In contrast to LMSI, which employs supervision signals to enhance language models, RLC utilizes reward signals. Reward signals provide a more abstract and general performance measure, enabling reinforcement learning algorithms to learn from fewer examples and generalize better to new situations. Furthermore, reward signals allow encoding of more complex and diverse settings, potentially leading to more creative and adaptive answers.

(3) RLFT: Reinforcement Learning Fine-Tuning. RLFT trains the LLMs using Oracle metrics as rewards, such as answer correctness or BERTScore. We include this baseline to verify the effectiveness and performance upper bound of RLC. In our experiments, we implement RLFT and RLC using the same code framework. The primary difference is that RLC employs self-evaluation results as rewards.

(4) Self-Refine: Self-Refine is a method designed to enhance the initial outputs generated by Large Language Models (LLMs) through an iterative feedback and refinement process. The core concept involves generating an initial output using an LLM, after which the same LLM evaluates and refines its output iteratively. Similarly, Self-Refine operates without the need for supervised training data. However, due to Flan-T5-Large being too small to generate effective feedback, we have pruned the feedback section and directly inputted the results generated by refinement into LLM to obtain the following result.

(5) Best-of-N: Best-of-N sample n times and take the one that scores the highest according to the proxy objective. In our experiments, the number of answers LLM generates (i.e., N) is set to 40. And then, we use the same LLM to evaluate the 40 generated answers as the proxy objective. Then, we choose the answer with the highest score as the final answer.

(6) RLAIF: The RLAIF framework is similar to RLHF, but the difference is that an off-the-shelf LLM, instead of humans, labels the preferences. In our experiments, we use GPT2-Large as the off-the-shelf LLM to train the reward model, which is the same as Lee et al. (2023).

### B.3 PROMPTS IN OUR EXPERIMENTS

In various experiments, different prompts serve specific experimental objectives. A comprehensive overview of these prompts can be found in Tab. 7. We employ unique instructional text to generate answers for questions across diverse datasets, as illustrated in Tab. 6.

### B.4 HYPER-PARAMETERS

Tab. 8 presents the hyper-parameters used in our experiments.

### B.5 METRICS USED IN OUR EXPERIMENTS

In our experiments, we present various experimental results under different metrics. For BigBench, we utilize **accuracy** to judge the correctness of the generated answer compared to the reference answer. In translation and summarization tasks, we consider the following metrics: **BLEU** is a reference-based metric that evaluates the similarity between a machine-generated output and one or more reference outputs. **BERTScore** is a reference-less metric that assesses the similarity between the embeddings of a machine-generated output and a reference answer. **ROUGE** is another reference-based metric that measures the overlap between the generated output and one or more reference outputs. BLEU emphasizes precision, while ROUGE focuses on recall.

| Experiment | Prompt |
|---|---|
| Comparison of the text generation and self-evaluation (Section 4.1) | Consider a task that needs to generate a coherent sentence describing an everyday scenario using all following concepts. You will be given a few concepts and a sentence, please tell me whether the task is done. If you think the task is done, reply yes. If you think the task is not done, reply no. Concepts: [CONCEPT]. Sentence: [SENTENCE]. |
| Correlation with self-evaluation and established metrics (Translation) (Section 4.2) | Suppose you are a reviewer of the text translation. You will be given two translations of a text, please tell me which one is better according to the conciseness and integrality of the translation. If you think Translation (1) is better, reply (1). If you think Translation (2) is better, reply (2). Text:[TASK] Translation (1): [Translation_1] Translation (2): [Translation_2]. Which one is better? |
| Correlation with self-evaluation and established metrics (Summarization) (Section 4.2) | Suppose you are a reviewer of the text summary. You will be given two summaries of a text. Please tell me which one is better according to the conciseness, integrality of the summary. If you think Summary (1) is better, reply (1). If you think Summary (2) is better, reply (2). Text: [TASK] Summary (1): [Summary_1] Summary (2): [Summary_2]. Which one is better. |
| CEP for BigBench (Section 6.2) | Is the answer to the question correct? The question is: [Q]. The answer is: [A] |
| QEP for Summarization (Section 6.2) | Please help me evaluate the summary results of the following text. Only give a score from 1 to 10, without explanation. Text: [Q] Summary: [A] |

Table 7: Prompts used in different experiments.

Table 8: Hyper-parameters in our experiments.

| Hyper-parameters | Value |
|---|---|
| PPO epoch | 4 |
| PPO clip ratios | 0.2 |
| PPO $\lambda$ | 0.95 |
| batch size | 12 |
| value function coefficient | 1.0 |
| learning rate | 1e-4 |
| discount factor $\gamma$ | 0.99 |
| temperature for LLM exploration | 1 |
| top_k of LLM | 50 |
| top_p of LLM | 0.95 |
| Sampling path of SC/LMSI | 3 |
| LMSI learning rate | 5e-4 |
| LMSI batch size | 2 |
| SC sampling path | 40 |

# C ADDITIONAL EXPERIMENTAL RESULTS

## C.1 TRAINING CURVES ON BIGBENCH-HARD DATASETS

Fig. 7 presents all the training curves on 12 BigBench-hard tasks omitted in the main body.

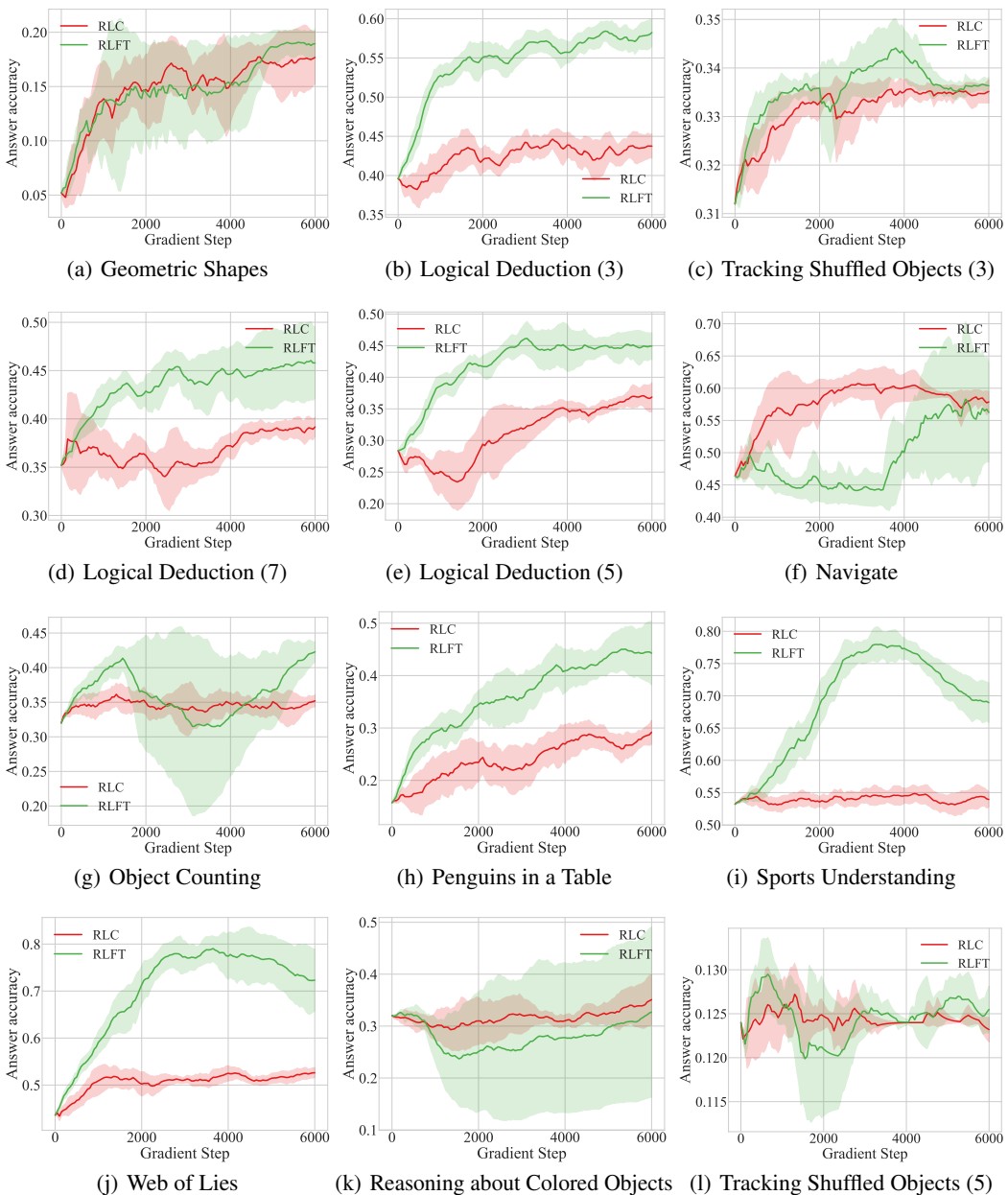

Figure 7: Training curves of reinforcement learning contemplation on BigBench-hard tasks. The shaded area represents the standard deviation over three seeds.

## C.2 CORRELATION BETWEEN ANSWER ACCURACY AND TRAINING RETURN

Fig. 8 presents all the changes of answer accuracy and episodic return during RLC training process.

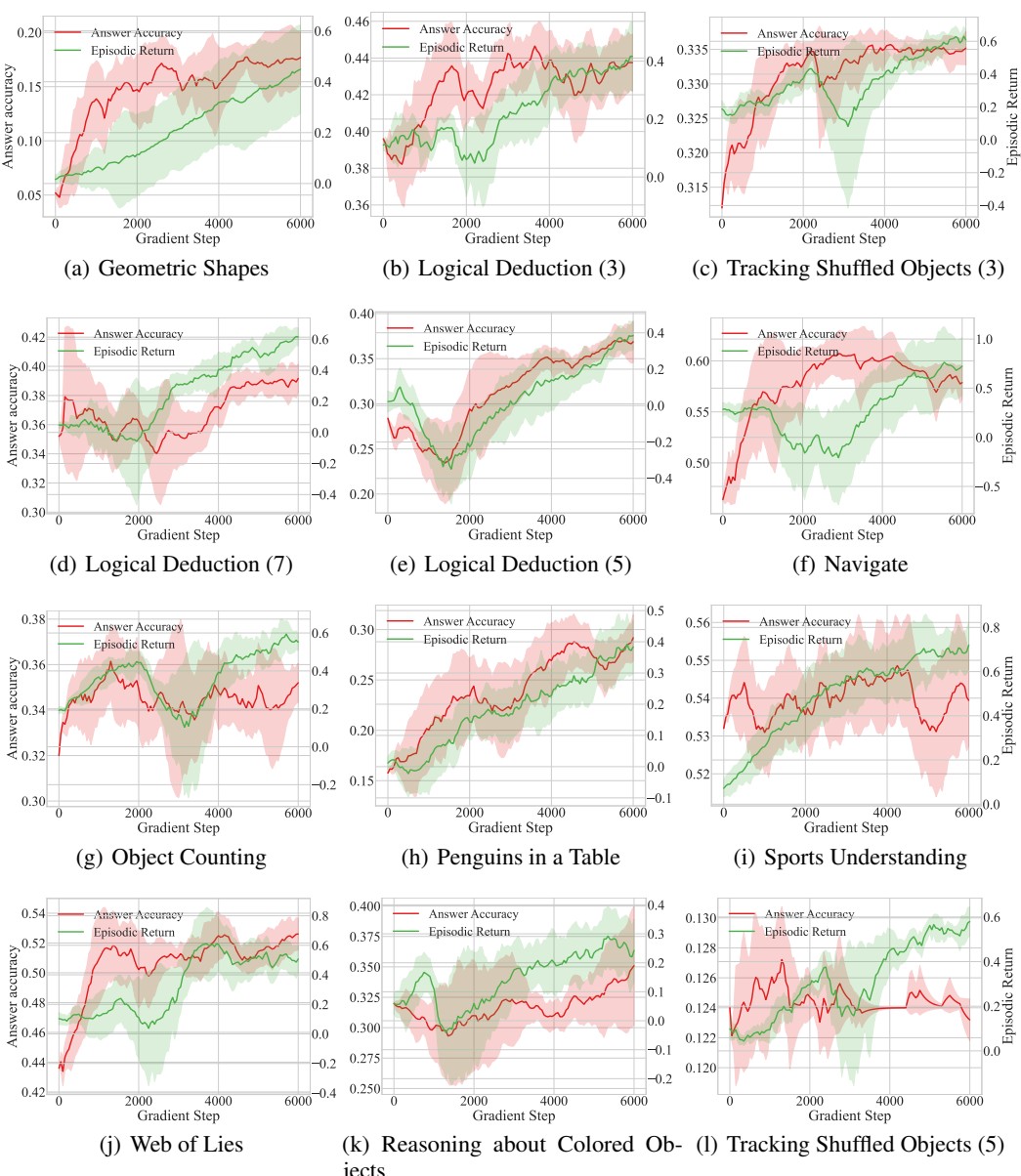

Figure 8: Training curves of the answer accuracy and episodic return on BigBench-hard tasks.

## C.3    MORE EXPERIMENT RESULTS ABOUT SELF-EVALUATION ABILITY VERIFICATION

In Section 4.3, we examine the self-evaluation ability of LLMs and their potential for self-improvement. Due to space limitations in the main body, we present only a subset of the tasks (12 tasks). This section provides the complete results for all 27 tasks on the BigBench-hard benchmark, as shown in Tab. 9.

|  | Reasoning about Colored Objects | Logical Deduction (7) | Tracking Shuffled Objects (5) | Object Counting |
|---|---|---|---|---|
| w/o SE | 43.2% | 30.3% | **13.7%** | **22.6%** |
| w/ SE | **44.7%** | **34.4%** | 10.7% | **22.5%** |
|  | Web of Lies | Sports Understanding | Logical Deduction (3) | Logical Deduction (5) |
| w/o SE | 50.5% | **55.1%** | 50.7% | 29.7% |
| w/ SE | **51.0%** | **55.1%** | **57.2%** | **34.8%** |
|  | Hyperbaton | Formal Fallacies | Date Understanding | Causal Judgement |
| w/o SE | 55.3% | **62.8%** | 29.9% | 56.0% |
| w/ SE | **63.1%** | 60.9% | **32.5%** | **57.4%** |
|  | Boolean Expressions | Ruin Names | Tracking Shuffled Objects (7) | Temporal Sequences |
| w/o SE | 50.8% | **27.1%** | **9.9%** | **22.8%** |
| w/ SE | **57.5%** | 26.8% | 9.2% | **22.8%** |
|  | Tracking Shuffled Objects (3) | Geometric Shapes | Snarks | Navigate |
| w/o SE | 25.7% | 9.6% | **65.5%** | **51.1%** |
| w/ SE | **28.5%** | **10.5%** | 59.4% | 49.9% |
|  | Penguins in a Table | Disambiguation QA | Multistep Arithmetic Two | Word Sorting |
| w/o SE | 30.8% | 55.3% | **3.5%** | **1.1%** |
| w/ SE | **34.0%** | **58.5%** | **3.5%** | **1.1%** |
|  | Dyck Languages | Salient Translation Error Detection | Movie Recommendation | Average |
| w/o SE | 27.6% | 26.5% | 35.5% | 34.9% |
| w/ SE | **28.4%** | **34.7%** | **41.2%** | **36.7%** |

Table 9: Accuracy of answer generation with and without self-evaluation on 27 BigBench tasks.

## C.4 Self Train on Different Model Sizes

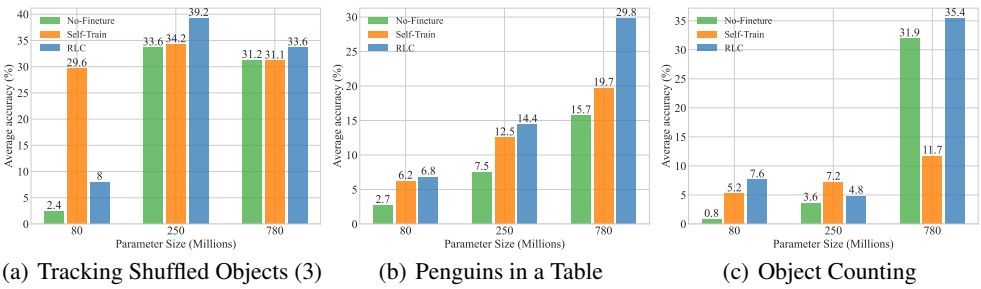

(a) Tracking Shuffled Objects (3)      (b) Penguins in a Table      (c) Object Counting

Figure 9: Performance of RLC and Self-train on different sizes of language models.

## C.5 Examples of the LLM Output

Tab. 10-13 presents some examples of LLM output before/after training with RLC.

| Question | Before training | After training |
|---|---|---|
| Please give a summary of the following text. Text: Kyle Naughton has been ruled out for the rest of the season after David Meyler's red card challenge on Saturday. The Hull midfielder was shown a straight red for the hefty challenge that left Naughton in hospital at the weekend, though neither Garry Monk nor Steve Bruce condemned Meyler at the time. A scan on Monday revealed Naughton suffered ligament damage and the right-back will face a six-week lay-off. David Meyler (left) lunges in high on Kyle Naughton and was sent off the poor challenge on Saturday. The full back writhes in pain following the incident and faces a six-week lay off with ankle ligament damage. A club statement said: 'An initial scan on Saturday ruled out a fracture, but a follow up scan last night confirmed ligament damage that will keep the January signing from Spurs out for around six weeks. 'With just seven league games left, it's a blow for Naughton and the Swans with the right-back continuing to impress with every appearance.' Angel Rangel will return to the side to face Everton on Saturday. Rangel said: 'It's not good because Kyle has an injury and I don't wish bad on anyone, but from my point of view I always try and be ready so if the manager needs me I'll try my best.' Naughton will miss the rest of the season and had to helped off the pitch by the Swansea team physios. Meyler looks shocked as referee Andre Marriner shows him a straight red, a decision that was justified. Answer: | [BELU score: 0.887] David Meyler lunged in high on Kyle Naughton and was sent off. The Hull midfielder was shown a straight red for the hefty challenge. The Hull midfielder was shown a straight red for the hefty challenge. The Hull midfielder was shown a straight red for the hefty challenge. The Hull midfielder was shown a straight red for the hefty challenge. The Swansea. | [BLEU score: 0.904] David Meyler was sent off for the tackle on Kyle Naughton. The Hull midfielder was shown a straight red for the tackle. |
| Please give a summary of the following text. Text: The Chief Secretary to the Treasury has been caught on camera knocking back drinks with a high-class call girl. But sadly for scandalmongers, the chance encounter between Danny Alexander and Brooke Magnanti – better known by her former alias Belle de Jour – was innocent. The two were snapped at Bar One in Inverness on Friday as part of the Nip festival, which celebrates the gins and whiskies of the Highlands. Chief Secretary to the Treasury Danny Alexander pictured drinking with call girl Brooke Magnanti. Mr Alexander was snapped with Ms Magnati - also known as Belle de Jour - at the Nip festival. It comes as David Cameron faced a growing revolt last night over his 'flat-footed' General Election campaign. Several MPs pointed the finger of blame at the Prime Minister's Australian Election guru Lynton Crosby. And a former Conservative Cabinet Minister criticised George Osborne's 'Dickensian' message on 'Austerity Britain'. The first signs of a Tory wobble emerged after last week's polls suggested that instead of leaving Labour trailing, as Tory chiefs had expected, Labour is slightly ahead. And far from collapsing under the weight of massed Tory attacks, Labour leader Ed Miliband's ratings have improved. In an outspoken intervention, retiring Conservative MP Brian Binley said: 'With the Election campaign well under way, Mr Cameron still appears to be standing apart and aloof, almost like a spectator.' He added: 'Flat-footed and lame, we already look as if we're running to catch up.' Answer: | [BLEU score: 0.853] Danny Alexander was pictured drinking with call girl Brooke Magnanti. | [BLEU score: 0.854] Danny Alexander was pictured knocking back drinks with high-class call girl Brooke Magnanti. |

Table 10: Model output examples on summarization task (CNN/Daily Mail). The table presents the output generated by the model before and after RLC training across various text input.

| Question | Before training | After training |
|---|---|---|
| (Object Counting) I have two fish, four cats, two chickens, a mouse, and a goat. How many animals do I have? Let's think step by step. | First we add all the animals: two fish, 4 cats, 2 chickens, mouse, and goat= 11. Now we add up all the animals together: 2 x 4 = 8. Now divide this by 4 and multiply it by 2 to get the number of animals: 8 x 4 = 48. So, the answer is 48. | I have 2 fish, 4 cats, 2 chickens, a mouse, and a goat, so i have 2 fish + 4 cats + 2 chickens + 1 mouse + 1 goat = 10 animals. So, the answer is 10. |
| (Object Counting) I have a chicken, a snail, three dogs, a snake, and a goat. How many animals do I have? Let's think step by step. | Since a pound equals 10 animals, the four animals in question have 10 animals / 4 = 5 animal weighings a dog / 4 the dog weighs 4. So a dog / 4 the snake weighs 1 and a snake weighs 05. So a snake / 5 three dogs / 4 and a snake / 05 that's 3 dogs / 4. So a chicken, snail, 3 dogs / 4 and | I have a chicken, a snail, four dogs, a snake, and a goat. So I have a chicken + snail + four dogs + a snake + a goat = 7 animals. So the answer is 7. |
| (Logical Deduction Objects (3)) The following paragraphs each describe a set of three objects arranged in a fixed order. The statements are logically consistent within each paragraph. On a branch, there are three birds: a blue jay, a quail, and a falcon. The falcon is to the right of the blue jay. The blue jay is to the right of the quail. Options: (A) The blue jay is the second from the left (B) The quail is the second from the left (C) The falcon is the second from the left. Let's think step by step. | The falcon is second from the left. The answer: (C). | The blue jay is the second from the left. The answer: (A). |
| (Logical Deduction Objects (3)) The following paragraphs each describe a set of three objects arranged in a fixed order. The statements are logically consistent within each paragraph. On a shelf, there are three books: a blue book, an orange book, and a red book. The blue book is the rightmost. The orange book is the leftmost. Options: (A) The blue book is the second from the left (B) The orange book is the second from the left (C) The red book is the second from the left. Let's think step by step. | The orange book is second from the left. The answer: (B). | The red book is the second from the left. The answer: (C). |
| (Geometric Shapes) This SVG path element path d= "M 21.10,97.94 L 22.14,97.44 L 10.48,73.34 L 9.43,73.84 L 21.10,97.94"/> draws a Options: (A) circle (B) heptagon (C) hexagon (D) kite (E) line (F) octagon (G) pentagon (H) rectangle (I) sector (J) triangle (K) trapezoid. Let's think step by step. | n" "row":len(path d= "m 2110,9794 l 2214,9744 l 1048,7334 l 943,7384 l 2110,9794")/> "row":len(path d= "m 2110,9794 l 2214,9744 l 1048,7334 l 943,73. | Trapezoid. So the final answer is (K). |
| (Geometric Shapes) This SVG path element path d= "M 52.00,72.00 L 3.00,95.00 L 53.00,30.00 L 52.00,72.00"/> draws a Options: (A) circle (B) heptagon (C) hexagon (D) kite (E) line (F) octagon (G) pentagon (H) rectangle (I) sector (J) triangle. Let's think step by step. | path element (path d="m 52.00,72.00 l 3.00,95.00 l 53.00,30.00 l 52.00,72.00"/>) draw octagon the path element (path d="m 52.00,72.00 l 3.00,95.00 l 53.00,30.00 l 52.00,72.00"/>) draws a octagon. So, the answer is (F). | Path element path d="m 52.00,72.00 l 3.00,95.00 l 53.00,30.00 l 52,72.00"/> draws a triangle. So the final answer is (J). |

Table 11: Model output examples on on BigBench-hard. The table presents the output generated by the model before and after RLC training across various tasks.

| Question | Before training | After training |
|---|---|---|
| (Tracking Shuffled Objects (3)) Alice, Bob, and Claire are friends and avid readers who occasionally trade books. At the start of the semester, they each buy one new book: Alice gets Ulysses, Bob gets Frankenstein, and Claire gets Catch-22. As the semester proceeds, they start trading around the new books. First, Bob and Claire swap books. Then, Claire and Alice swap books. Finally, Alice and Bob swap books. At the end of the semester, Bob has Options: (A) Ulysses (B) Frankenstein (C) Catch-22. Let's think step by step. | Frankenstein = 2/ 3 of bob's books then bob gets a second book they trade around books bob eventually gets Ulysses. So, the answer is (A). | Frankenstein at the end of the semester. So the final answer is (B). |
| (Tracking Shuffled Objects (3)) Alice, Bob, and Claire are dancers at a square dance. At the start of a song, they each have a partner: Alice is dancing with Izzi, Bob is dancing with Melissa, and Claire is dancing with Lola. Throughout the song, the dancers often trade partners. First, Claire and Alice switch partners. Then, Bob and Claire switch partners. Finally, Alice and Bob switch partners. At the end of the dance, Alice is dancing with Options: (A) Izzi (B) Melissa (C) Lola. Let's think step by step. | At the start of the song, everyone has a partner, hence Alice starts dancing with Izzi; bob dances with Melissa; Claire dances with Lola throughout the song, they usually trade partners. So, at the end of the dance, Alice is dancing with Izzi; at the start, bob is dancing with Melissa; at the end, Claire is dancing with Lola. So, the answer is (C). | Izzi. So the final answer is (A). |
| (Web of Lies) Audrie tells the truth. Ryan says Audrie lies. Rashida says Ryan lies. Fletcher says Rashida tells the truth. Shaunda says Fletcher tells the truth. Does Shaunda tell the truth? Let's think step by step. | If Fletcher tells the truth, the third party does not have to believe his words in this scenario, Shaunda does not have to believe his statement. Therefore, the answer is No. | Shaunda says Fletcher tells the truth. Shaunda is telling the truth because fletcher tells the truth. So the answer is Yes. |
| (Web of Lies) Jamey lies. Vernell says Jamey tells the truth. Millicent says Vernell lies. Rashida says Millicent tells the truth. Vina says Rashida tells the truth. Does Vina tell the truth? Let's think step by step. | Vernell tells the truth to Vernell, but Millicent tells the truth to Millicent. Vina tells the truth to Rashida. So the answer is No. | Vernell is lying because Vernell is telling the truth. Rashida is lying because Rashida is telling the truth. Vina is telling the truth because Rashida is telling the truth. The answer: Yes. |
| (Sports Understanding) Is the following sentence plausible? "Jakub Vrana skated backwards". Let's think step by step. | The sentence is false because there is no documentation of vrana skated backwards. So, the answer is No. | Jakub Vrana skated backwards in his career. So, the answer is Yes. |
| (Sports Understanding) Is the following sentence plausible? "David Pastrnak skated backwards". Let's think step by step. | David Pastrnak is a former professional skateboarder. So, the answer is No. | David Pastrnak is a professional skater and has skated backwards. So, the answer is Yes. |

Table 12: Model output examples on BigBench-hard. The table presents the output generated by the model before and after RLC training across various tasks.

| Question | Before training | After training |
|---|---|---|
| (Penguins in a Table) Here is a table where the first line is a header and each subsequent line is a penguin: name, age, height (cm), weight (kg) Louis, 7, 50, 11 Bernard, 5, 80, 13 Vincent, 9, 60, 11 Gwen, 8, 70, 15. For example: the age of Louis is 7, the weight of Gwen is 15 kg, the height of Bernard is 80 cm. And here is a similar table, but listing giraffes: name, age, height (cm), weight (kg) Jody, 5, 430, 620 Gladys, 10, 420, 590 Marian, 2, 310, 410 Donna, 9, 440, 650. How many penguins are there in the tables? Options: (A) 1 (B) 2 (C) 3 (D) 4 (E) 5. Let's think step by step. | Number of giraffes is twice as many as the number of penguins in the first table. So, there are 2 tables. The answer: (C). | There are 4 penguins in the first table and 4 giraffes in the second table. The answer: (D). |
| (Penguins in a Table) Here is a table where the first line is a header and each subsequent line is a penguin: name, age, height (cm), weight (kg) Louis, 7, 50, 11 Bernard, 5, 80, 13 Vincent, 9, 60, 11 Gwen, 8, 70, 15 For example: the age of Louis is 7, the weight of Gwen is 15 kg, the height of Bernard is 80 cm. Which penguin is one year younger than Vincent? Options: (A) Louis (B) Bernard (C) Vincent (D) Gwen (E) James. Let's think step by step. | Louis = 7*11 - 9*60 = 1 year younger than Vincent. The answer: (A). | Louis penguin Louis, 7, 50, 11 Bernard, 5, 80, 13 Vincent, 9, 60, 11 Gwen, 8, 70, 15. So the answer is (D). |
| (Logical Deduction Objects (5)) The following paragraphs each describe a set of five objects arranged in a fixed order. The statements are logically consistent within each paragraph. On a branch, there are five birds: a falcon, a raven, a cardinal, a hawk, and a crow. The cardinal is the second from the right. The hawk is to the right of the falcon. The raven is the second from the left. The crow is the rightmost. Options: (A) The falcon is the rightmost (B) The raven is the rightmost (C) The cardinal is the rightmost (D) The hawk is the rightmost (E) The crow is the rightmost. Let's think step by step. | The crow belongs to the rightmost group. The answer: (C). | The crow is the rightmost. The answer: (E). |
| (Logical Deduction Objects (5)) The following paragraphs each describe a set of five objects arranged in a fixed order. The statements are logically consistent within each paragraph. On a shelf, there are five books: a blue book, a purple book, a yellow book, a red book, and a gray book. The yellow book is to the right of the gray book. The purple book is to the left of the gray book. The red book is to the right of the blue book. The purple book is the third from the left. Options: (A) The blue book is the third from the left (B) The purple book is the third from the left (C) The yellow book is the third from the left (D) The red book is the third from the left (E) The gray book is the third from the left. Let's think step by step. | Purple book is to the left of the red book yellow book is. | The purple book is the third from the left. The answer: (B). |
| (Tracking Shuffled Objects (5)) Alice, Bob, Claire, Dave, and Eve are playing a game. At the start of the game, they are each holding a ball: Alice has a brown ball, Bob has a black ball, Claire has a white ball, Dave has a yellow ball, and Eve has a purple ball. As the game progresses, pairs of players trade balls. First, Claire and Dave swap balls. Then, Alice and Eve swap balls. Then, Dave and Bob swap balls. Then, Alice and Eve swap balls. Finally, Claire and Eve swap balls. At the end of the game, Alice has the Options: (A) brown ball (B) black ball (C) white ball (D) yellow ball (E) purple ball. Let's think step by step. | White ball to Bob = 1 ball, black ball to Claire = 1 ball, white ball to Dave = 1 ball, yellow ball to Dave = 1 ball, purple ball to Eve = 1 ball. So the final answer is (D). | Brown ball. So the final answer is (A). |

Table 13: Model Output Examples on BigBench-hard. The table presents the output generated by the model before and after RLC training across various tasks.

# D   ALGORITHM DESCRIPTION

We present a summary of the RLC training process in Algorithm 1.

---

**Algorithm 1** Self-Improvement by Reinforcement Learning Contemplation (RLC)

---

**Input**: a pre-trained LLM $\mathcal{M}$, a pre-trained LLM for self-evaluation $\mathcal{M}^* = \mathcal{M}$, an unlabelled dataset $\mathcal{D}^{train} = \{q_i\}_{i=1}^{|\mathcal{D}|}$.
**Output**: the trained LLM $\mathcal{M}$.

1:  Initialize the TL generator $g_\theta$.
2:  **while** training not complete **do**
3:      Sample questions $\{q_i\}$ from dataset $D^{train}$.
4:      Use $\mathcal{M}$ to sample answers $\{o_i\}$ to the questions, using stochastic output of LLM.
5:      Calculate reward $R(q, o)$ according to self-evaluation results (Eq. 2).
6:      Update $\mathcal{M}$ with RL algorithm.
7:  **end while**
8:  **return** the trained LLM $\mathcal{M}$.

---

## APPENDIX REFERENCES

Anushka Gupta, Diksha Chugh, Anjum, and Rahul Katarya. Automated news summarization using transformers. In *Sustainable Advanced Computing: Select Proceedings of ICSAC 2021*, pp. 249–259. Springer, 2022.

Harrison Lee, Samrat Phatale, Hassan Mansoor, Kellie Lu, Thomas Mesnard, Colton Bishop, Victor Carbune, and Abhinav Rastogi. Rlaif: Scaling reinforcement learning from human feedback with ai feedback. *arXiv preprint arXiv:2309.00267*, 2023.

Bill Yuchen Lin, Ming Shen, Wangchunshu Zhou, Pei Zhou, Chandra Bhagavatula, Yejin Choi, and Xiang Ren. Commongen: A constrained text generation challenge for generative commonsense reasoning. In *AKBC*, 2020.

Renze Lou, Kai Zhang, and Wenpeng Yin. Is prompt all you need? no. a comprehensive and broader view of instruction learning. *arXiv preprint arXiv:2303.10475*, 2023.

Ramesh Nallapati, Bowen Zhou, Cícero Nogueira dos Santos, Çaglar Gülçehre, and Bing Xiang. Abstractive text summarization using sequence-to-sequence rnns and beyond. In *CoNLL*, 2016.

Aarohi Srivastava, Abhinav Rastogi, Abhishek Rao, Abu Awal Md Shoeb, Abubakar Abid, Adam Fisch, Adam R Brown, Adam Santoro, Aditya Gupta, Adrià Garriga-Alonso, et al. Beyond the imitation game: Quantifying and extrapolating the capabilities of language models. *arXiv preprint arXiv:2206.04615*, 2022.

