# OpenReview forum: "Language Model Self-improvement by Reinforcement Learning Contemplation"
_ICLR.cc/2024/Conference — ICLR 2024 poster_

### Official Review · Reviewer_jzQi · 2023-10-15

**Soundness:** 3 good
**Presentation:** 4 excellent
**Contribution:** 2 fair
**Rating:** 5
**Confidence:** 4

**Summary:**

The authors present a method Reinforcement Learning Contemplation (RLC) as an alternative to existing RLAIF workflows. Rather than training a preference model + doing PPO using the preference model, they just directly prompt the original model for reward signal, following analysis indicating that evaluation is easier than generation. They evaluate on a number of natural language reasoning tasks and outperform their baselines.

**Strengths:**

--Comprehensive results on several reasoning tasks, with interesting analyses.

--Interesting to see that it seems to work reasonably at small model scales, unlike the original Anthropic RLAIF paper

--The new pipeline is a bit simpler than RLAIF, not needing to train a preference model as an intermediate step, while still working well

**Weaknesses:**

--I'm not sure that the idea of evaluation being easier than generation is particularly new; you could view this as the intuition for RLAIF too, where you get the model to evaluate itself and then use the preference model to improve its generations later using PPO. I'm not sure it's the right motivation for your work anyway; if I understood correctly, the main difference between your method and RLAIF is that you *evaluate the absolute quality of only one output* at a time (either binary or 1-10), rather than *evaluating the relative quality of two outputs*. Then naturally you don't bother fitting a preference model since you already have the reward due to predicting absolute quality rather than relative quality.

--On a related note, these tasks seem a bit nonstandard for RLAIF evaluation, are there other works using these too? I'm curious whether the results hold up when you do the classic tasks like harmlessness and helpfulness. In particular, if I understand correctly, one major difference is that the tasks you evaluate seem to have a "gold" "correct" answer, unlike e.g. harmlessness and helpfulness, which makes it reasonable for your model to label the absolute quality of a given output rather than comparing relative quality between two outputs.

**Questions:**

--Do the RLAIF and other baselines also get to use chain of thought for the reasoning tasks, like your method?

--I saw you mentioned in the appendix that you use GPT2-Large for the reward model for RLAIF, rather than the same Flan-T5 model you use for the main model. Is it fair to use Flan-T5 to evaluate outputs for your model, while using finetuned GPT2-Large (presumably weaker) to compare outputs for your baseline?

--What are the prompts used for RLAIF / other baselines?

--Using language models to directly output 1-10 scores seems like it might suffer from poor calibration - why not e.g., try to finetune with a regression head on top rather than just using the LM out of the box? Would that improve performance?

--The numbers in 6.4 don't seem very convincing for what you're claiming, if anything it seems like the performance went up more on the unrelated tasks (object counting, penguins)?

--Nit: I don't really see why you distinguish between RLCAI and RLAIF; from what I can tell you basically mean the same approach for both terms? (And the paper you cite for RLAIF attributes the term RLAIF to the original Anthropic paper anyway.)

---

> ### Author Response · Authors · 2023-11-18
> **Author response (Part 1/2)**
>
> We are grateful for the time and effort you put into reviewing our work. Below we address each of your concerns and questions.
>
> > Q1: I'm not sure that the idea of evaluation being easier than generation is particularly new.
>
> It is an intuitive idea that evaluation is easier than generation. Thus some previous works may use this idea more or less. They commonly use the evaluation ability of LLM heuristically (e.g., RLAIF, Self-Refine, etc.). In contrast, our study offers a thorough experimental validation of the performance disparity between evaluation and generation capabilities in language models.
>
> > Q2: Regarding the BigBench-Hard datasets used in the experiment.
>
> **Comment 1: On a related note, these tasks seem a bit nonstandard for RLAIF evaluation.**
>
> **Response 1:** The key insight of RLC is “LLM self-improve by contemplation”, which can help LLM to improve continuously on the given tasks. As for RLAIF, There is also a lot of work to use text generation tasks([1], [2], [3], [4]) such as summarization and reasoning tasks such as BBH ([1], [2]).
>
> **Comment 2: I'm curious whether the results hold up when you do the classic tasks like harmlessness and helpfulness.**
>
> **Response 2:** Good point. We have conducted additional experiments employing RLC to train FLAN-T5-Large in generating negative sentences. Specifically, we provide language models with a continuation prompt: "Continue with a story starting with 'I just went to the cinema'". Subsequently, the model's output is evaluated as positive, neutral, or negative, with corresponding rewards of 0, 1, and 2, respectively. We collected 200 examples of model output and subjected them to human evaluation. The results indicate that 92% of the generated sentences were negative, demonstrating the effectiveness of RLC in producing the desired output. This suggests that the approach can be extended to other tasks, such as harmlessness and helpfulness, to further validate its efficacy.
>
> | Method | Answer accuracy |
> | ------ | --------------- |
> | RLC    | 92%             |
>
> > Q3: Is it fair to use Flan-T5 to evaluate outputs for your model, while using finetuned GPT2-Large (presumably weaker) to compare outputs for your baseline?
>
> We use GPT2-Large due to it has similar size (774M) as FLAN-T5-Large (780M), and GPT series models have been applied as reward by ChatGPT. However, we recognize that FLAN-T5-Large could be more proper thus we conduct additional experiments that replace GPT2-Large with FLAN-T5-Large. The experimental results are shown in the following table. The experiment results show that there is no significant change in the performance of RLAIF (FLAN-T5-Large). Of course, it's better than RLAIF (GPT-2 Large).
>
> |                               | RLC   | RLAIF (GPT-2 Large) | RLAIF (FLAN-T5-Large) |
> | ----------------------------- | ----- | ------------------- | --------------------- |
> | Object Counting               | 35.4% | 32.5%               | 34.7%                 |
> | Penguins in a Table           | 29.8% | 19.8%               | 24.9%                 |
> | Tracking Shuffled Objects (3) | 33.6% | 32.8%               | 32.4%                 |
>
> > Q4: Using language models to directly output 1-10 scores seems like it might suffer from poor calibration. Why not e.g., try to finetune with a regression head on top rather than just using the LM out of the box? Would that improve performance?
>
> Implementing fine-tuning with a regression head appears promising for evaluating text quality. However, this approach necessitates supervised data for effective fine-tuning, while this paper mainly focuses on improving language models without external supervision.
>
> To address potential calibration issues, a possible solution is to reduce the evaluation score range. This adjustment aims to enable the language model to more accurately assess text quality.
>
> > Q5: The numbers in 6.4 don't seem very convincing for what you're claiming, if anything it seems like the performance went up more on the unrelated tasks (object counting, penguins)?
>
> Apologize for any confusion caused by the presentation in the original version. The increase in performance is not necessarily correlated with the similarity between the training tasks and the evaluation tasks. For instance, Logical Deduction (3) (+1.5) and Tracking Shuffled Objects (5) (-0.1) both have tasks akin to the training tasks. It is important to note that the tasks in question were not seen during the training process. Despite this, the RLC method demonstrates an overall improvement in performance on average. We will update the paper to rectify any misleading presentation and provide clearer explanations.

---

> ### Author Response · Authors · 2023-11-18
> **Author response (Part 2/2)**
>
> > Q6: Regarding questions about the experiments.
>
> **Comment 1: Do the RLAIF and other baselines also get to use chain of thought for the reasoning tasks, like your method?**
>
> **Response 1:** Yes. All baselines in our experiments also use CoT prompt: “Let’s think step by step”.
>
> **Comment 2: What are the prompts used for RLAIF / other baselines?**
>
> **Response 2:** We use the same prompts for all baselines and RLC. The prompts we used in the experiments are listed in Table 6 in Appendix for the details of prompts.
>
> | BigBench-hard      | [TEXT] Let’s think step by step.                          |
> | ------------------ | --------------------------------------------------------- |
> | Text summarization | Please give a summary of the following text. Text: [TEXT] |
> | Answer             |                                                           |
>
> > Q8: I don't really see why you distinguish between RLCAI and RLAIF.**
>
> We simply use two terms to indicate they are from different works. We mainly use the term ‘RLAIF’ in the whole paper.
>
> ## Reference：
>
> [1] Sun Z, et al. Salmon: Self-alignment with principle-following reward models, 2023.
>
> [2] Yang C, et al. Large language models as optimizers[J]. arXiv preprint arXiv:2309.03409, 2023.
>
> [3] Bai Y, et al. Constitutional ai: Harmlessness from ai feedback, 2022.
>
> [4] Lee H, et al. RLAIF: Scaling reinforcement learning from human feedback with ai feedback, 2023.
>
> ------
>
> We hope that these responses can address your concerns and questions. If you had any further concerns, please let us know.

---

> ### Author Response · Authors · 2023-11-23
>
> Hi reviewer jzQi. As the discussion period comes to an end, we want to follow up to see if the response addresses your concerns. If you have any further questions, please let us know. Thank you again!

---

### Official Review · Reviewer_Rzu4 · 2023-10-16

**Soundness:** 3 good
**Presentation:** 4 excellent
**Contribution:** 3 good
**Rating:** 8
**Confidence:** 4

**Summary:**

The authors introduce a self-improvement method for language models, i.e. a method that enables a language model to improve without any external supervision, as opposed to RLHF. It builds on the intuition that it is easier for language models to evaluate text than to generate it. In section 4, the authors confirm this intuition quantitatively on experiments from standard benchmarks and show that text evaluation can enable non-invasive improvement of language models by post-processing of text generations.
They introduce a self-evaluating task-specific reward model (RLC) for “invasive” improvement of the LMs. Similarly to RLAIF (which asks the LM to rank generated text), it tailors a prompt for each task, possibly giving a denser reward signal to PPO: a strong case for RLC is it does not require learning a reward model. Empirical results on a large number of tasks show that RLC performs better than best-of-N, RLHF, plain RLAIF and other baselines to fine-tune small models.

**Strengths:**

- The paper is very well structured and well written. I particularly liked the structure and factorization of the appendix, as well as the experiments from Section 4 that confirms the intuition that LMs are better at evaluation (as humans)
- Showing that the internal reward / ranking capabilities of LMs are correlated with heuristics metrics (BLEU, BERTScore, ROUGE).
- Experiments are well-executed and done on a variety of standard tasks.

**Weaknesses:**

**Weaknesses**
- Section 4.3 looks a bit weak to me (greedy and 3 samples) to evaluate the accuracy of best-of-N with the best evaluated text versus accuracy of average of LM samples. Furthermore, greedy is evaluated against 1 or 2 samples)
- Why doing only 3 samples (this experiment does not require any fine-tuning and is done on a small LM whose identity is unknown).
- The method is simple and not novel: it relies on two “template” reward prompts (defined section 5) and good parsing of LM’s output (as opposed to RLAIF that learns a reward model). This is not a problem to me as long as experiments are done rigorously.

**Additional experiments  (could make me increase a bit my score)**
- Could you plot the evaluation of the LM when fine-tuning from RLC (for instance on the Figure 7 from appendix), as well as the heuristic metrics (BLEU, BERTScore, ROUGE) when applicable?
- Would appreciate a discussion (backed by experiments) on the RLC/RLAIF updates compared to doing more sampling and non-invasive self-evaluation.

**Recommendations (could make me increase my score considerably):**
- From “self”-improvement to “any”-improvement: what if the evaluation model is different from the LM to improve?
would be interesting to have a more complete view: e.g. using smaller/bigger evaluation models. Similarly to the RLAIF baseline that uses GPT2 through the RLAIF baseline in 6.2 but not using RLC.
- The authors mentioned two reward prompts (section 5): it could be interesting to make the LM learn search prompts given a task description.

**Questions:**

- It is not clear to me how accuracy is computed in experiments of section 4.1.
- What is the difference between best-of-N and w/ SE in section 4.3?
- Do you have an intuition why preference-based RLAIF is weaker than RLC? Better “reward” prompts in RLC as opposed to the use preferences in RLAIF? Having to explicitly learn a reward model?
- In Table 4, any idea why RLFT is stronger than RLC on “Reasoning about Colored Objects”?
- It is often not really clear what model is used (4.2, 4.3, 6) and why results are not reported for different model sizes (as claimed in the abstract)


Minor nitpicks:
- Define Self-Evaluation (SE) in 4.3
- Define Self-consistency (SC) before self-train as SC is mentioned before definition.

---

> ### Author Response · Authors · 2023-11-18
> **Author response (Part 1/2)**
>
> Thank you for your careful review and comprehensive comments. Please find our response below.
>
> > Q1: Section 4.3 looks a bit weak to me & Why doing only 3 samples？
>
> We would like to clarify that the experiment in Section 4.3 aims to verify the potential of text evaluation to improve text generation. We focus more on investigating whether the LM can improve when equipped with self-evaluation mechanism.
>
> Consequently, we employ a straightforward approach in which the language model generates output while simultaneously evaluating its performance. The experimental results, based on three samples, indicate that language models exhibit improvement when utilizing self-evaluation, thus supporting the central motivation of this paper.
>
> > Q2: The novelty of the method.
>
> We acknowledge the prior work, RLAIF, shares the same setting with RLC. However, we would like to clarify their differences in motivation and framework. the main point of our work is to explain why a kind of RLAIF functions effectively. We provide evidence that evaluation tasks are simpler than generation tasks for LLMs. This difference can be leveraged to boost the performance of LLMs of various sizes.
>
> In contrast, prior RLAIF studies relied mostly on comprehensive experiment results, some of which necessitated the LLMs to be robust enough. The experimental findings (referenced in Table 4) illustrate this particular limitation.
>
> Furthermore, RLC offers a more straightforward implementation for LMSI, eliminating the need for gathering preference data and training a reward model.
>
> > Q3: Regarding suggestions about additional experiments.
>
> Following your suggestion, we have conducted additional experiments.
>
> **Comment 1: would be interesting to have a more complete view: e.g. using smaller/bigger evaluation models. Similarly to the RLAIF baseline that uses GPT2 through the RLAIF baseline in 6.2 but not using RLC.**
>
> **Response 1:** Thank you for your suggestion. It’s a great direction for RLC to explore. We have conducted supplementary experiments employing FLAN-T5 models of different sizes as evaluation models, specifically FLAN-T5-Small (80M), FLAN-T5-Base (250M), and FLAN-T5-Large (780M). The results demonstrate that, generally, larger evaluation models yield improved performance. Notably, even the smaller evaluation model, FLAN-T5-Small, delivers satisfactory results, supporting the claim that evaluation is a less complex task.
>
> |                                         | DG    | RLC-S | RLC-B | RLC-L |
> | --------------------------------------- | ----- | ----- | ----- | ----- |
> | object_counting                         | 31.9% | 31.1% | 35.5% | 35.4% |
> | penguins_in_a_table                     | 15.7% | 28.8% | 27.4% | 29.8% |
> | tracking_shuffled_objects_three_objects | 31.2% | 34.5% | 32.3% | 33.6% |
>
> **Comment 2: Could you plot the evaluation of the LM when fine-tuning from RLC (for instance on the Figure 7 from appendix), as well as the heuristic metrics (BLEU, BERTScore, ROUGE) when applicable?**
>
> **Response 2:** Good point. This result would be helpful to understand what happens during the RLC training process. However, these heuristic metrics (BLEU, BERTScore, and ROUGE) are not applicable to BigBench-Hard tasks as these metrics cannot be directly calculated. As supplementary, we plot the episodic return curves during the training process, as shown in Figure 8 in Appendix in the revised version of manuscript. The experiment results show that the answer accuracy has a consistent trend as the episodic return, demonstrating the effectiveness of the self-evaluation.

---

> ### Author Response · Authors · 2023-11-18
> **Author response (Part 2/2)**
>
> > Q4: Would appreciate a discussion (backed by experiments) on the RLC/RLAIF updates compared to doing more sampling and non-invasive self-evaluation.
>
> Thank you for your suggestion. Both RLC and RLAIF are invasive LMSI methods. In contrast, non-invasive methods, such as Self-Consistency, do not require additional training for language models. This distinction results in different trade-offs between these approaches.
>
> Language models trained using invasive methods like RLC and RLAIF can generate responses more rapidly, making them suitable for applications with stringent response time requirements, such as chatbots. However, these methods necessitate additional training, which increases computational costs and complexity.
>
> On the other hand, non-invasive methods do not require further training of the language models, reducing computational costs and simplifying their implementation. While these methods may not be as fast in generating responses, they offer a more resource-efficient alternative.
>
> We suggest that both invasive and non-invasive LMSI methods are closely related to the concept of **autonomous agents**, which must solve tasks and continuously improve. These two categories of methods can naturally cater to these essential capabilities. We will incorporate this discussion, along with relevant experimental results, at the end of Section 6 (Experiments) to provide a comprehensive comparison of these approaches.
>
> > Q5: It is not clear to me how accuracy is computed in experiments of section 4.1.
>
> Apologize for any ambiguity in our initial presentation.
>
> **To assess the accuracy of text generation**, we use human evaluation. A generated text is considered "correct" if it inclusively incorporates all the concepts presented in the question.
>
> **To assess the accuracy of text evaluation**, we first prompt the same LLM to self-evaluate if the generated text contains all the specified concepts, utilizing the prompt given in Table 7. Then we mark the evaluation result as “correct” if the LLM's evaluation result is same to that of human.
>
> > Q6: What is the difference between best-of-N and w/ SE in section 4.3?
>
> In Section 4.3, we present two distinct implementations of the W/ SE method for the BigBench-Hard and summarization tasks. For the BigBench-Hard task, the W/ SE approach involves regenerating an answer if the language model evaluates the previous response as incorrect, which is different from best-of-N (i.e., choose best one from N samples). In contrast, for summarization tasks, W/ SE generates three potential answers and selects the optimal one based on evaluation results. This latter approach is similar to the best-of-N method.
>
> > Q7: Discussions about the experiment results.
>
> **Comment 1: Do you have an intuition why preference-based RLAIF is weaker than RLC?**
>
> **Response 1:** A key distinction between RLAIF and RLC lies in the method of obtaining rewards. In RLC, rewards are directly derived from self-evaluation outcomes, whereas RLAIF relies on rewards trained from preference data labeled by the language model. This difference in reward acquisition may contribute to the weaker performance of RLAIF compared to RLC.
>
> **Comment 2: any idea why RLC is stronger than RLFT on “Reasoning about Colored Objects”?**
>
> **Response 2:** The RL tasks in our experiments are characterized by sparse reward problems, where agents receive a binary 0-1 reward signal upon generating complete answers. This may make it challenging for language models to learn from sparse reward signals in certain tasks. As observed in Figure 7 of the Appendix, RLC and RLFT exhibit similar training curves. A potential solution to address this issue is to replace the binary reward with a softer one, such as using the language model's probability of evaluating an answer as correct or incorrect as the reward.
>
> > Q8: It is often not really clear what model is used (4.2, 4.3, 6) and why results are not reported for different model sizes.
>
> In Sections 4.3 and 6, we employ the Flan-T5-Large model, while for Section 4.2, we utilize the Flan-T5-XL model. Our primary objective in Sections 4.2 and 4.3 is to validate the hypothesis that "language model can self-improve using it evaluation ability". These experiments in Section 4 only serve as a validation step, rather than the main results of the paper. Therefore, we have not conducted experiments with varying model sizes. For experiments in Section 6, we have conducted experiments with variety of FLAN-T5 model sizes, as the results shown in Figure 6.
>
> ------
>
> We hope that our response has addressed your concern and questions satisfactorily. If you had any further concerns, we are glad for discussion.

---

> > ### Comment · Reviewer_Rzu4 · 2023-11-19
> > **Final remarks**
> >
> > Thanks to the authors for the strong rebuttal.  I will update my score positively, but have a last question related to your answer to my question on "a discussion (backed by experiments) on the RLC/RLAIF updates compared to doing more sampling and non-invasive self-evaluation.".
> >
> > My main concern is that evaluation could be unfair between direct generation (DG) / best-of-N with a fixed (and small) number of model generations and self-improving models (both invasive and non-invasive). Indeed, I would expect the latter to be evaluated with more generations/samples.  Selecting the top candidate from a larger pool of generations could artificially make results look better, so having a fixed number of generations for all methods would probably be a more fair comparison. Do you have any insights on this? Thanks.

---

> ### Author Response · Authors · 2023-11-19
> **Response to follow-up comment**
>
> Thanks for your timely response. We are glad to find our response has addressed your proposed issues.
>
> We understand your concerns about the sample generation number of LMSI methods. While we recognize that employing multiple sample generations could potentially enhance the performance of LMSI methods, it's essential to highlight that our experimental focus lies in assessing the efficacy of various methods in enhancing language models. Our evaluation primarily measures the performance disparity between DG (which involves a single sampling). Consequently, in our experiments, LMSI methods are restricted to a single sampling instance.
>
>
> However, we agree that conducting experiments combining LMSI methods with multiple sampling instances would be valuable. This approach could effectively showcase performance enhancements over these varied sampling methodologies, such as 'best-of-N', thereby enriching the experimental outcomes. We assure you that the experiments will be conducted and integrated into the future version of the paper.
>
>
> ———
>
> If you had any further questions, we are glad to discuss.

---

> > ### Comment · Reviewer_Rzu4 · 2023-11-19
> > **Last clarification**
> >
> > Thanks for your clarification. "Consequently, in our experiments, LMSI methods are restricted to a single sampling instance." -> I could be missing something here but the RL process of RLC involves multiples samples. Can you elaborate on the selection of the reported sample?

---

> > > ### Author Response · Authors · 2023-11-19
> > >
> > > We appreciate your follow-up comments.
> > >
> > > Sorry for the maybe ambiguous presentation.
> > > "Restricted to a single sampling instance" here means that the language model (of RLC, RLAIF, RLFT, DG, self-train) only generates one answer as the final answer during the answer generation process. On the other hand, some of the methods (e.g., SC, best-of-N) generate multiple answers and select one as the final answer from the answer candidates. During the training process, RLC does collect multiple samples for RL training. The confusion may result from the usage of the improper presentation "single sampling instance".

---

> > > > ### Comment · Reviewer_Rzu4 · 2023-11-20
> > > > **Last last clarification**
> > > >
> > > > Thanks for your answer. Considering the rebuttal, I am raising my score to 8.

---

> > > > > ### Author Response · Authors · 2023-11-21
> > > > >
> > > > > We appreciate your thorough review and are pleased to have addressed your concerns. Thank you for your valuable time and effort in evaluating this paper.

---

### Official Review · Reviewer_BB78 · 2023-10-31

**Soundness:** 2 fair
**Presentation:** 2 fair
**Contribution:** 2 fair
**Rating:** 3
**Confidence:** 3

**Summary:**

This work proposes a method for training language models without the need for external supervision or additional reward models. First, the language model is used for generation. Second, the language model is used to evaluate its own generated outputs. Third, the evaluation result is used to improve the generation. This approach results in some gains on BigBench tasks.

**Strengths:**

This work demonstrates empirical gains using self-improvement via evaluating self-generated outputs.

**Weaknesses:**

1. The approaches demonstrated in this work is not new, and the authors do not discuss prior work. For there is a family of work in evaluating self-generation (https://arxiv.org/abs/2210.03629), how is this work different?
2. This work uses CoT, do the baselines use CoT as well? How much of the gain is coming from CoT? This work is missing ablations that quantify the gains from the primary contribution.
3. The majority of this manuscript describe background information. The main contribution of this work starts on page 5. I suggest that the authors drastically truncate the background content and use the remaining room to analyze the primary contribution. First, how does it compare to prior work? What is generated during self-evaluation? Where do they agree/disagree with reference evaluation?

**Questions:**

What is the statistical significance of the gains in Tables 2 and 4? Several of the columns show what seems to be minor gains (e.g. for Table 2, all except Penguins in a Table). Are these gains within standard deviation?

---

> ### Author Response · Authors · 2023-11-18
> **Author response (Part 1/2)**
>
> Thank you for your time and valuable comments. We have taken every comment into consideration. Please find the response below.
>
> > Q1: For there is a family of work in evaluating self-generation (https://arxiv.org/abs/2210.03629), how is this work different?
>
> The work [2] you mentioned and its related works ([1, 3]) lie in the field of language model self-correction. The primary differences between our work and these previous works can be summarized as follows:
>
> 1. **Motivation:** We would like to highlight that the main point of our work is to explain why a kind of RLAIF functions effectively. We provide evidence that evaluation tasks are simpler than generation tasks for LLMs. This difference can be leveraged to boost the performance of LLMs of various sizes. Meanwhile, previous studies relied mostly on comprehensive experiment results, some of which necessitated the LLMs to be solid enough.
> 2. **Implementation:** These previous works refining their output during the generation process by repeated self-evaluation and revision. In contrast, invasive methods update LM’s parameters to improve its capability.
> 3. **Inference speed:** RLC offers a direct generation process after training, contrasting with these previous works, which entails slow generation due to multiple rounds of evaluation and revision.
> 4. **Method justification:** We justify the performance gap between text evaluation and generation through comprehensive experiments in Section 4, while previous works typically utilize the evaluation ability of LLM heuristically.
>
> We would like to discuss and highlight these differences in Section 2 (Related Work).
>
> > Q2: This work uses CoT, do the baselines use CoT as well? How much of the gain is coming from CoT? This work is missing ablations that quantify the gains from the primary contribution.
>
> In our experiments, we utilize the CoT prompt [4], "Let's think step by step," for all baseline models. Consequently, the performance gain of RLC can be assessed by comparing its answer accuracy to that of the DG method. We apologize for any ambiguity in the original manuscript. To address this, we will revise Section 6.1 (Experimental Setup) to provide a clearer explanation of the experimental design.
>
> > Q3: Regarding the paper organization.
>
> We believe there could be some misunderstanding about the paper's organization. Please find our response below.
>
> **Comment 1: The main contribution of this work starts on page 5.**
>
> **Response 1:** Sorry for the potential misleading about the paper organization. We would like to clarify that Section 4 (starting from page 3) presents an important contribution of this paper: the verification of the performance gap between text generation and evaluation. **This performance gap serves as the foundation of the design of RLC.** Meanwhile, we have some discussions about the potential of language model self-improvement leveraging such a gap.
>
> **Comment 2: I suggest that the authors drastically truncate the background content and use the remaining room to analyze the primary contribution.**
>
> **Response 2:** Thanks for your suggestions. We would like to simplify the related work section and add more discussions about the differences between this work and previous works. Besides, we will update Section 5 (Method) to discuss how RLC utilizes self-evaluation, an its differences in comparison to prior works.

---

> ### Author Response · Authors · 2023-11-18
> **Author response (Part 2/2)**
>
> > Q4: What is the statistical significance of the gains in Tables 2 and 4?
>
> We understand your concern that the improvement in Table 2/4 is not significant. We would like to clarify the experiment results as follow:
>
> 1. Concerns about the significance of the gains.
>
>    It is important to note that Reasoning dataset is inherently challenging for language model enhancement. We specifically utilized the "Hard" subset within the Reasoning task. For BigBench-Hard [5] dataset, an advanced model from OpenAI, text-avinci-001, fail to answer with a high accuracy [5] on Tracking Shuffled Objects. Moreover,  some previous methods exhibit poor improvements in reasoning tasks. For example, self-refine [1] exhibits 0% and 0.2% enhancements with GPT3.5 and GPT4 model, respectively. Furthermore, ReAct [2] demonstrates performance degradation on HotpotQA (a popular reasoning task).
>
> 2. Distinction between Tables 2 and 4:
>
>    Table 2 demonstrates the potential of self-evaluation in improving text generation without incorporating the proposed method. It reveals that a language model can improve its performance using a self-evaluation mechanism. In contrast, Table 4 presents the results of the proposed RLC method, which surpasses the  DG method on 11 out of 12 tasks.  These findings substantiate the effectiveness of the RLC method.
>
> 3. Regarding the standard deviation: To ensure the robustness of our experimental setup, we conducted all experiments using three distinct seeds, thereby reducing the impact of deviations.
>
> ## References:
>
> [1] Madaan A, et al. Self-Refine: Iterative refinement with self-feedback, 2023.
>
> [2] Yao S, et al. ReAct: Synergizing reasoning and acting in language models, 2023.
>
> [3] Shinn N, et al. Reflexion: an autonomous agent with dynamic memory and self-reflection, 2023.
>
> [4] Wei J, et al. Chain-of-thought prompting elicits reasoning in large language models, 2022.
>
> [5] Suzgun M, et al. Challenging big-bench tasks and whether chain-of-thought can solve them, 2022.
>
> ------
>
> We hope the response has addressed your concerns. But if you have any further questions, please let us know.

---

> ### Author Response · Authors · 2023-11-23
>
> Hi reviewer BB78. We wanted to follow up to see if the response addresses your concerns. If you have any further questions, please let us know. Thank you again!

---

### Official Review · Reviewer_ksaG · 2023-11-01

**Soundness:** 3 good
**Presentation:** 3 good
**Contribution:** 3 good
**Rating:** 8
**Confidence:** 3

**Summary:**

This paper proposes an approach for language model self-improvement: given an LM, improve it using itself to provide feedback. This paper proposes a method called Reinforcement Learning Contemplation (RLC) to do this, based on the observation that LMs are better at providing feedback versus generating full examples.

The three stage process is to 1. sample a bunch of QA pairs from the model (the questions come from a training dataset), 2. evaluate the QA pair from the model and transform it into a reward, and then 3. update the base LM accordingly.

The paper evaluates this over CommonGen (lexically constrained text generation), 12 reasoning tasks from BigBench-Hard, along with CNN-daily mail summarization. The experiments show improvements over baselines like sampling once or N times, as well as self-training approaches without RL. Training on seed data from multiple tasks also shows some generalization to unseen tasks.

**Strengths:**

Overall, this paper seems reasonably solid to this reviewer. The approach proposed is simple and general, yet it also seems novel and underexplored at least to this reviewer. The results suggest that it helps on reasoning tasks which could suggest something interesting is happening during the RL process. It's good to see results on a variety of different model sizes too, which suggest also that the gain doesn't go away with scale (at least for e.g. the BigBench task "Penguins in a Table")

I have some concerns about evaluation below though I think/hope they can be answered in rebuttal --

**Weaknesses:**

The main concerns to this reviewer are:
* the datasets considered are a bit toy. It would be great to see other experiments from other domains (maybe something like math word problems?) or things considered by some of the other common papers in this space.
* The approach is limited to having a training dataset, though this is addressed appropriately in Appendix A.1 (at least to this reviewer, maybe extending this could be left for future work). However, I am a bit concerned that there might be stronger baselines that make use of that unlabeled training dataset.
* It's not clear to this reviewer what the model is learning here -- is it learning to truly reason or just to produce better answers (e.g. that mimic the format of the seed dataset)? It would be great to run more controlled experiments studying this. The main comparison that seems to address this is the "Self-train" baseline that generates high-confidence answers and then trains on them. It would be great to have a version of Figure 6 (performance of RLC across sizes of LMs) comparing with "Self-train" instead of with no finetuning.

**Questions:**

For the experiments, how do the non-PPO trained models generate answers / what happens if they generate an answer or assign high probability mass outside of the set of candidate options (e.g. for MultiChoice answering something that's not A/B/C/D/E?

---

> ### Author Response · Authors · 2023-11-18
> **Author response**
>
> Thanks for your positive comments and constructive feedback on the paper. Below we address each of your concerns and questions.
>
> > Q1: The datasets considered are a bit toy. It would be great to see other experiments from other domains (maybe something like math word problems?)
>
> Thanks for your suggestion. In our experiments, we primarily employed the BigBench-Hard and CNN-Daily-Mail datasets, which are commonly utilized to assess various capabilities of LLM. To address your concern, we have incorporated an additional dataset, MathQA [1], renowned for its diverse and challenging problem set. The experimental results, as presented in the table below, demonstrate that RLC improves answer accuracy from 14.9% to 19% on the MathQA dataset. This finding suggests that RLC can be effectively applied to other datasets, thereby showcasing its versatility and applicability.
>
> | Method | Accuracy |
> | ------ | -------- |
> | RLC    | 19%      |
> | DG     | 14.8%    |
>
> > Q2: The approach is limited to having a training dataset …
>
> **Comment 1: limited to have a training dataset**
>
> **Response 1**: In this work, we consider the language model self-improvement with a training dataset, which is a common setting in LMSI studies. We are encouraged that you are also interested in training the language models without a dataset, which is also the ultimate goal of RLC. To address this, we have conducted supplementary experiments aimed at improving language models without relying on a dataset. Specifically, we provide language models with a continuation prompt: "Continue with a story starting with 'I just went to the cinema'". Subsequently, the model's output is evaluated as positive, neutral, or negative, with corresponding rewards of 0, 1, and 2, respectively. We collected 200 examples of model output and subjected them to human evaluation. The results indicate that 92% of the generated sentences were negative, demonstrating the effectiveness of RLC in producing the desired output. This suggests that the approach can be extended to other tasks, such as harmlessness and helpfulness, to further validate its efficacy.
>
> | Method | Output accuracy |
> | ------ | --------------- |
> | RLC    | 92%             |
>
> **Comment 2: I am a bit concerned that there might be stronger baselines that make use of that unlabeled training dataset**
>
> **Response 2**: To the best of our knowledge, the baselines in our experiments are representative methods in the domain , encompassing invasive methods (such as self-training, RLAIF, Best-of-N) and non-invasive methods (SC, Self-Refine), etc. We remain open to conducting further experiments with any additional baselines that demonstrate proficiency in LMSI.
>
> > Q3: It's not clear to this reviewer what the model is learning here & It would be great to have a version of Figure 6 (performance of RLC across sizes of LMs) comparing with "Self-train" instead of with no finetuning.
>
> We agree that utilizing the Self-train method as a baseline can provide valuable insights into the learning process of the language model. The Self-train method aims to improve the language model's ability to generate accurate answers.
>
> As per your recommendation, we have updated Figure 6 (please refer to the revised version), adding Self-train method. We observe that RLC consistently outperforms the Self-train method across various language model sizes in terms of answer accuracy.
>
> This results indicate the language model could learn new knowledge during the reinforcement learning contemplation process, in addition to the accurate answer.
>
> > Q4: For the experiments, how do the non-PPO trained models generate answers…
>
> In the experiments, all models generate answers using an auto-regressive approach to generate token sequences. These models are not restricted to outputting valid options, which is consistent with the experimental settings employed in prior research [2]. Consequently, language models may produce answers that fall outside the set of candidate options, which would be evaluated as ‘wrong’ during the evaluation process.
>
> It is important to note that, PPO-trained models (RLC, RLAIF, RLFT) could generate the answer out of the candidate options, as these models also output the token sequence, rather than the discrete actions like in traditional RL.
>
> ## References above:
>
> [1] Amini A, Gabriel S, et al. MathQA: Towards interpretable math word problem solving with operation-based formalisms, 2019.
>
> [2] https://github.com/FranxYao/chain-of-thought-hub/tree/main/BBH
>
> ------
>
> We hope that these responses can address your concerns and questions. If you had any further concerns, please let us know.

---

> > ### Comment · Reviewer_ksaG · 2023-11-23
> > **thanks! keeping my score**
> >
> > thanks for the author response! I think these address my key concerns so I vote to keep my score and accept this paper. thanks!

---

> > > ### Author Response · Authors · 2023-11-23
> > >
> > > We are glad that our response has addressed your key concerns. Thanks for providing valuable suggestions in improving this paper.

---

### Official Review · Reviewer_scBc · 2023-11-01

**Soundness:** 3 good
**Presentation:** 3 good
**Contribution:** 2 fair
**Rating:** 6
**Confidence:** 4

**Summary:**

This paper presents a method for improving a language model using its own output. Specifically, the authors first show that there is an accuracy gap in the output of a language model between generation and evaluation, using Bigbench and summarization tasks. They then present a method for fine-tuning the language model using the evaluation output as the reward in reinforcement learning. The authors use Flan-T5 (780M) to demonstrate the effectiveness of their proposed approach.

**Strengths:**

- The fact that evaluation is sometimes easier than generation is well known, but the authors show concrete experimental results that support this in non-trivial settings.
- Overall, the paper is well written and easy to follow.

**Weaknesses:**

- The novelty of the work is limited. The overall approach is very similar to RLAIF.
- The effectiveness of the proposed approach is demonstrated using the 780M model of Flan-T5, but it is not clear how effective it is when other or larger models are used.

**Questions:**

- My understanding is that Flan-T5 is an encoder-decoder model, which may have helped to boost the accuracy of evaluation since the encoder transformer allows each token to attend to any other token in the input. I am wondering if the big accuracy gap between  generation and accuracy still exists in decoder-only models.
- Figure 2 suggests that the accuracy gap between generation and evaluation becomes smaller as the model size increases. Is the proposed approach still effective for larger models?
- Is it true that GPT-2 Large was used in Lee et al. (2023)?  It seems to me that they used PaLM 2.

---

> ### Author Response · Authors · 2023-11-18
> **Author response**
>
> Thank you for carefully reviewing our paper and providing constructive comments. We hope that our response has addressed your concerns, but if we missed anything please let us know.
>
> > Q1: The novelty of the work is limited.
>
> We would like to highlight that the main point of our work is to explain why a kind of RLAIF functions effectively. We provide evidence that evaluation tasks are simpler than generation tasks for LLMs. This difference can be leveraged to boost the performance of LLMs of various sizes. Meanwhile, prior RLAIF studies relied mostly on comprehensive experiment results, some of which necessitated the LLMs to be robust.
>
> > Q2: The effectiveness of the proposed approach is demonstrated using the 780M model of Flan-T5 &  Is the proposed approach still effective for larger models?
>
> We appreciate your suggestions and have conducted experiments on FLAN-T5-XL (3B), which is about 4 times larger than FLAN-T5-Large (780M). The experiment results are presented in the following table. We observe that RLC can be applied to larger language model. As future work, we will try to conduct experiments with larger model (e.g., FLAN-T5-XXL (11B), LLaMA2, etc).
>
>    | Dataset                       | DG    | RLC (FLAN-T5-Large)   |
> | ----------------------------- | ----- | ----- |
> | Penguins in a Table           | 33.8% | 39.2% |
> | Tracking Shuffled Objects (3) | 24.0% | 27.4% |
> | Object Counting               | 23.4% | 26.6% |
>
> > Q3: I am wondering if the big accuracy gap between generation and accuracy still exists in decoder-only models.
>
> This is an interesting question. In the development of the method, RLC does not impose specific requirements on the model architecture. This versatility allows for the exploration of the accuracy gap in various models, including decoder-only configurations.
>
> > Q4: Figure 2 suggests that the accuracy gap between generation and evaluation becomes smaller as the model size increases.
>
> As discussed in Section 4.1, evaluation accuracy is influenced by both the evaluation capability and the quality of the generated text. As the model size increases, the language model can produce higher quality text that becomes more challenging to differentiate. Consequently, the gap may narrow when the language model exhibits exceptional text generation abilities. We posit that the experimental results in Section 4.1 provide an intuitive representation of the performance disparity between text evaluation and text generation.
>
> > Q5: Is it true that GPT-2 Large was used in Lee et al. (2023)? It seems to me that they used PaLM 2.
>
> Sorry for the unclear presentation regarding the GPT-2 Large model of RLAIF. It is true that Lee et al. employed the PaLM 2 model as the LM labeler. However, the size of the model is positively correlated with its capabilities, and if the model is too large (such as PaLM 2), it is difficult to compare whether it is the improvement brought about by capabilities or methods. Therefore, in order to ensure fairness in comparison, in our study, we utilize the GPT-2 Large (774M) model as both the labeler and reward model, which is comparable in size to the FLAN-T5-Large (780M) model.
>
> Furthermore, to further verify the impact of the reward model, we conducted supplementary experiments in which we replaced the GPT-2 Large model with the FLAN-T5-Large model, aligning our implementation with the RLC that employs FLAN-T5-Large as the evaluator. The results of these experiments are presented in the table below. We can observe that RLAIF shows close performance on different model. In the future, we will also try to conduct experiments with different types of models for RLAIF and RLC to compare their performance.
>
>
> |                                   | RLC   | RLAIF (GPT-2 Large) | RLAIF (FLAN-T5-Large) |
> | --------------------------------- | ----- | ------------------ | -------------------- |
> | **Object Counting**               | **35.4%** | 32.5%              | 34.7%                |
> | **Penguins in a Table**           |**29.8%** | 19.8%              | 24.9%                |
> | **Tracking Shuffled Objects (3)** | **33.6%** | 32.8%              | 32.4%                |
>
> ------
>
> Thanks again for the careful and timely response. We are glad to any further discussions.

---

> ### Author Response · Authors · 2023-11-21
> **Supplementary results on decoder-only models**
>
> Dear reviewer scBc,
>
> Regarding your concerns about (Q3) *if the big accuracy gap between generation and accuracy still exists in decoder-only models*, we have conducted supplementary experiments using the widely-recognized decoder-only model, ChatGPT (GPT-3.5-turbo), on the CommonGen task. These experiments encompass varying levels of difficulty, as determined by the number of concepts involved. The updated experimental results are presented below:
>
> | Concept Num.    | 6        | 8        | 10       | 11       | 12      | 13       | 14       | 15       | 16       |
> | --------------- | -------- | -------- | -------- | -------- | ------- | -------- | -------- | -------- | -------- |
> | Generation Acc. | **0.78** | **0.72** | 0.61     | 0.51     | 0.49    | 0.43     | 0.41     | 0.36     | 0.23     |
> | Evaluation Acc. | 0.72     | 0.64     | **0.62** | **0.53** | **0.60** | **0.51** | **0.42** | **0.42** | **0.29** |
>
> The experiment results indicate that evaluation accuracy generally surpasses generation accuracy. When the generation task is relatively simple (concept number ≤ 8), the language model can achieve high generation accuracy, resulting in a slightly higher generation accuracy compared to evaluation accuracy. However, as the generation task becomes more complex, a performance gap between generation and evaluation emerges. This experiment demonstrates that the accuracy gap persists in decoder-only models. We would like to conduct further investigation with additional decoder-only models to provide a more comprehensive results.

---

> > ### Comment · Reviewer_scBc · 2023-11-22
> >
> > Thank you for the response and revision. Most of my concerns have been addressed and I've updated my score.
> >
> > It would be nice if the paper could include a brief discussion of encoder-decoder models vs decoder-only models in terms of the proposed approach.

---

> > > ### Author Response · Authors · 2023-11-22
> > >
> > > We are glad that our response has addressed most of your concerns and appreciate your updating score.
> > >
> > > > It would be nice if the paper could include a brief discussion of encoder-decoder models vs decoder-only models in terms of the proposed approach.
> > >
> > > RLC mainly utilizes language models evaluate the text or answer the questions, which necessitate the models' capabilities to follow the instructions provided by prompts. Decoder-only models are known for their ability to follow prompts effectively. Conversely, encoder-decoder models are commonly utilized for complex tasks like summarization or translation.
> > > However, we did not conduct experiments with decoder-only models like LLaMA (7B/13B/70B), due to the computation limitation. Instead, we employed FLAN-T5, which has been fine-tuned with a focus on following the instruction, in our experiments. It would be interesting to apply RLC to decoder-only models, including LLaMA and the GPT series.
> > >
> > > ------
> > >
> > > Thanks for your suggestions. We would like to add these discussions in the Experiment section.

---

### Author Response · Authors · 2023-11-21
**General response & Paper update**

## General response

We express our gratitude to the reviewers and chairs for their valuable time and constructive feedback on our paper. We have carefully considered each comment and provided detailed responses. We would like to highlight that the contribution (reviewer scBc, BB78, Rzu4) of this work is to explain why a kind of RLAIF functions effectively. We provide evidence that evaluation tasks are simpler than generation tasks for LLMs. This difference can be leveraged to boost the performance of LLMs of various sizes. Meanwhile, prior RLAIF studies relied mostly on comprehensive experiment results, some of which necessitated the LLMs to be robust enough.

We have conducted additional experiments to support the motivation and the effectiveness of the proposed method. We demontrate the performance gap between evaluation and generation on ChatGPT (reviewer scBc), evaluate RLC on new datasets (Reviewer ksaG), provide more results of baseline methods (Reviewer scBc, jzQi, ksaG), present results with larger language models (Reviewer scBc) and different evaluation models (Reviewer scBc), and demonstrate the training process of RLC (reviewer Rzu4).

With respect to the performance improvements (Reviewer BB78) presented in Table 2, we wish to clarify that these results are intended to confirm the feasibility of employing text evaluation to refine text generation capabilities. They are preliminary verifications rather than the main results of the RLC. Please refer to our individual response for more details.

## Paper update

We have made significant improvements to our paper based on the valuable feedback provided by the reviewers. We highlight the revised content in *blue* in the **[updated version](https://openreview.net/pdf?id=38E4yUbrgr)** for your convenience. We believe that the paper is now more polished and easier to understand. Here are the main updates:

1. Additional experiments have been conducted to resolve the concerns raised, with Figures 6 and 7 updated and new Figures 8 and 9 introduced to reflect these changes. (Reviewers scBc, jzQi, ksaG, Rzu4)
2. We have reorganized Section 1 (Introduction) to better clarify our contributions and novelty. (Reviewer scBc, BB78, Rzu4)
3. We update/truncate Section 2 to add more discussions about previous works in evaluating self-generation. (Reviewer BB78)
4. Section 5 (Method) has been updated to elaborate on how RLC incorporates self-evaluation and its difference compared to previous methods. (Reviewer BB78)
5. We add a discussion about the answer generation processes of different methods in Section 6.1 (Experiment Setup), and clarify the fairness of the comparison. (Reviewer Rzu4)
6. Section 6.4 has been revised to elucidate the experimental results pertaining to the *generalization on unseen tasks*. (Reviewer jzQi)
7. The experimental setup, including prompts, is now more comprehensively detailed in Section 6.1 (Experiment Setup). (Reviewers jzQi, BB78)

---

### Author Response · Authors · 2023-11-22
**Requesting feedback on responses**

Dear reviewers,

We appreciate the time and effort you have dedicated to reviewing our paper and offering detailed feedback. As the discussion period draws to a close, we kindly request confirmation on whether any issues or questions persist that require further clarification. We are open to and welcome additional discussions regarding our work.


Best regards,
Submission 52 authors.

---

### Meta-Review · Area_Chair_kLtJ · 2023-12-13

**Metareview:**

The authors present a method Reinforcement Learning Contemplation (RLC) as an alternative to existing RLAIF workflows. Rather than training and using the preference model, they just directly prompt the original model for reward signal, following analysis indicating that evaluation is easier than generation. They evaluate on a number of natural language reasoning tasks and outperform baselines.

Reviewers liked that the method is general and the experiments are convincing. They also liked how the method works on smaller models in contrast to anthropic. The main objection is about novelty where verification being easier than generation is not new or is perhaps obvious. This has been addressed well, and clearly testing "obvious" hypothesis is an important part of science. Other objections include the need for unlabeled training data, using atypical tasks for the setting, and that some of the evaluations are toy, and alternative evaluations. The authors have good responses to these points in the discussions.

The main negative review (BB78) had some misunderstandings about the paper, which were addressed in the original paper already and further clarified in author response. The reviewer did not respond further on these points.

1) most importantly their work as an invasive method that modifies the model which is big difference compared to the reference pointed out by the reviewer.

2) The baselines also use CoT in the paper, and the authors promised to further clarify to avoid ambiguity

3) It seems fair to say that clearly identifying an issue and providing experimental results is a contribution over just believing it to be the case.

**Justification For Why Not Higher Score:**

reviews are a bit mixed

**Justification For Why Not Lower Score:**

Good paper that provides a clear analysis and a solution. Simpler than existing approaches and generally applicable.

---

### Decision · Program_Chairs · 2024-01-16

Accept (poster)